# TE-VLM: Transfer Entropy for Vision Language Model Distillation

## Abstract

Vision-Language Models (VLMs) have demonstrated impressive performance across various multimodal tasks. However, deploying large teacher models in real-world applications is often infeasible due to their high computational cost. To address this, knowledge distillation has been widely explored to transfer knowledge from a large teacher model to a smaller student model. In this paper, we propose a novel distillation framework that integrates Transfer Entropy (TE) as a regularization term to enhance information flow from the teacher to the student model. TE quantifies the directional dependency between teacher and student embeddings, encouraging the student model to effectively capture structural knowledge from the teacher. To efficiently approximate TE in high-dimensional embedding spaces, we introduce two surrogate formulations based on cosine similarity: (1) TE via cosine similarity of directional changes in embeddings and (2) TE via concatenated differences across modalities. Our experiments, conducted on the MSCOCO 2014, Flickr8k, Flickr30k, Food-101, and ImageNet-1k datasets using CLIP-based teacher and student architectures, demonstrate that incorporating TE significantly improves retrieval performance. Through extensive analysis, we show that TE-based regularization enhances the student model's ability to capture multimodal associations and maintain representational consistency. Our findings suggest that TE is an effective tool for improving knowledge transfer in VLM distillation, bridging the performance gap between compact student models and their larger teacher counterparts.

## 1 Introduction

Vision-Language Models (VLMs) have emerged as a powerful framework for learning joint representations of images and text, enabling applications such as image captioning, visual question answering, and cross-modal retrieval (Radford et al., 2021; Jia et al., 2021). However, state-of-the-art VLMs are often computationally expensive, making them impractical for deployment in resource-constrained environments. To address this challenge, knowledge distillation (Hinton et al., 2015) has been widely adopted to transfer knowledge from a large teacher model to a smaller, more efficient student model while maintaining performance.

Existing approaches to VLM distillation primarily rely on contrastive learning (Li et al., 2022; Yang et al., 2024) and divergence-based losses, such as Kullback-Leibler (KL) divergence (Li et al., 2024b), to align the student model's probability distribution with that of the teacher. However, these methods do not explicitly quantify the directed information flow between the teacher and student representations. As a result, traditional distillation losses may overlook the sequential and structural dependencies present in the learning dynamics of multimodal embeddings.

To overcome these limitations, we propose a novel distillation framework that integrates Transfer Entropy (TE) as a regularization mechanism to enhance the student model's ability to mimic the teacher's information transfer patterns. TE is a measure of directed information flow between two systems, originally developed in information theory (Schreiber, 2000). In the context of VLM distillation, TE quantifies how much additional knowledge the teacher provides to the student beyond what the student has already learned from past

states. This allows us to explicitly encourage meaningful knowledge transfer, ensuring that the student model captures the evolving structure of the teacher's representations.

The key contributions of this work are as follows:

- We introduce TE as a regularization method for VLM distillation, explicitly capturing the directed information flow from the teacher model to the student.

- We theoretically show that the first-order (linear) expansion of TE leads to a computable surrogate based on a cosine similarity between the teacher and student-process Jacobians.

- We propose two novel TE approximations based on cosine similarity, enabling efficient computation of information transfer in high-dimensional multimodal embeddings.

- We demonstrate that integrating TE into the distillation loss function leads to significant improvements in retrieval performance, outperforming traditional contrastive, KL-divergence, Mean Squared Error (MSE), Interactive Contrastive Learning (ICL), and Mutual Information (MI) distillation methods.

- We provide extensive empirical validation on the MSCOCO 2014, Flickr8k, Flickr30k, Food-101, and ImageNet-1k datasets using different teacher-student distillation setups, showing that TE-based regularization enhances multimodal representation learning and improves student model alignment with the teacher.

## 2 Related Work

### 2.1 Knowledge Distillation

Knowledge distillation enables the transfer of learned representations from a large teacher network to a smaller student model (Hinton et al., 2015). Building on this idea, intermediate feature supervision has been proposed to guide deeper yet more efficient networks (Romero et al., 2015). Other approaches emphasize spatial attention alignment (Zagoruyko & Komodakis, 2017), or address scenarios where original training data is unavailable through data-free distillation (Huang & Wang, 2017).

Further research has focused on aligning internal representations between teacher and student networks. Contrastive objectives harmonize feature spaces (Tian et al., 2020), while attention-based strategies have been tailored for transformer architectures (Touvron et al., 2021). Information-theoretic methods have also been explored, including maximizing mutual information (Ahn et al., 2019) and modeling inter-sample relationships (Park et al., 2019). Another line of work distills the teacher's probability distribution to improve student generalization (Passalis & Tefas, 2018). Mutual-relation distillation techniques have also been applied to face recognition, as demonstrated in CoupleFace (Liu et al., 2022). In multimodal representation learning, Chen et al. (2023) proposed an objective that preserves mutual information between a teacher model and an auxiliary modality to improve distillation.

More recently, several studies have examined frequency-domain representations. Frequency attention modules enable students to adapt feature responses under teacher guidance (Pham et al., 2024), semantic frequency prompts have been leveraged for dense prediction tasks (Zhang et al., 2024), and frequency-aware optimization has also been used to construct compact synthetic datasets (Shin et al., 2023).

Other relevant contributions include self-distillation for general-purpose text embeddings (Chen et al., 2024), sample-efficient distillation through synthetic training data (Liu et al., 2024), dual-teacher frameworks (Li et al., 2024c), orthogonal-projection-based transfer (Miles et al., 2024), and automated search strategies for distillation pipelines in object detection (Li et al., 2024a).

### 2.2 Vision–Language Model Distillation

In the vision–language domain, early work aligned object semantics with textual descriptions to improve cross-modal understanding (Li et al., 2020), while large-scale pre-training frameworks such as UNITER

learned universal image–text representations through joint optimization (Chen et al., 2020). More recently, research has increasingly focused on compressing and distilling large vision–language models to obtain efficient students while preserving strong multimodal alignment. For example, Fang et al. (2021) explored model compression for multimodal encoders, Dai et al. (2022) distilled vision–language knowledge to enable multimodal generation, and Wang et al. (2022) refined lightweight CLIP-style models for downstream tasks. TinyCLIP (Wu et al., 2023) further demonstrated that affinity mimicking and weight inheritance can effectively shrink CLIP models while maintaining competitive zero-shot performance.

Distillation has also been leveraged to enhance multimodal reasoning and transferability. Instruction-tuning frameworks distill higher-level reasoning ability from strong multimodal teachers (Hu et al., 2024), and multimodal knowledge transfer has been applied to enable open-vocabulary object recognition (Gu et al., 2021). In parallel, frequency-aware distillation strategies have been proposed to improve robustness under distribution shift (Li et al., 2023), and vision–language representations have been used to strengthen classification across diverse domains (Addepalli et al., 2024).

Very recent work continues to expand the scope and structure of VLM distillation. Mixture-of-Visual-Encoder Knowledge Distillation (MoVE–KD) aggregates complementary information from multiple teacher encoders into a single student model (Cao et al., 2025), while Align–KD encourages students to capture cross-modal interactions earlier in the representation hierarchy (Feng et al., 2025). Several objective functions have also been proposed specifically for CLIP-style distillation: Yang et al. (2024) introduced interactive contrastive learning and MSE alignment, and Li et al. (2024b) employed KL-based distribution matching. Our work is complementary to these formulations, but differs in that we introduce a transfer entropy–based reward signal designed to guide and regularize cross-modal knowledge transfer.

## 3 Introduction to Transfer Entropy

Transfer Entropy is an information-theoretic measure introduced by Schreiber (Schreiber, 2000) to quantify the directed transfer of information between two stochastic processes. It is particularly useful for detecting asymmetrical interactions and causal relationships, as it measures the influence that the past of one process, $X$, has on the future of another process, $Y$, beyond what can be explained by the past of $Y$ alone.

For two discrete-time stochastic processes $X(t)$ and $Y(t)$, the transfer entropy from $X$ to $Y$ is formally defined as (Schreiber, 2000):

$$T_{X \to Y} = \sum_t p(y_{t+1}, y_t, x_t) \log \frac{p(y_{t+1} \mid y_t, x_t)}{p(y_{t+1} \mid y_t)}, \tag{1}$$

where $p(\cdot)$ represents probability distributions of the respective random variables.

Transfer entropy is closely related to conditional mutual information. It can be rewritten as the conditional mutual information between $Y_{t+1}$ and $X_t$, conditioned on $Y_t$ (Shahsavari Baboukani et al., 2020):

$$T_{X \to Y} = I(Y_{t+1}; X_t \mid Y_t), \tag{2}$$

where $I(Y_{t+1}; X_t \mid Y_t)$ is the mutual information between $Y_{t+1}$ and the history of $X$, conditioned on the history of $Y$. The proof is provided in Appendix A. This formulation reveals that transfer entropy measures the additional information that $X_t$ provides about the future state $Y_{t+1}$, over and above the information provided by $Y$'s own history $Y_t$. In Appendix B, we present an overview of prior work on mutual information and TE estimation.

## 4 Method

### 4.1 Why Is Transfer Entropy Beneficial for VLM Distillation

In VLM distillation, the goal is not only to match the teacher's representations, but to ensure that the teacher's signals *drive* the student's updates in both modalities. Transfer Entropy (TE) provides an

information-theoretic way to conceptually quantify this update-driving effect by measuring how much the teacher explains the student's *next-step* representations beyond what is already explained by the student's current state.

**Notation.** For a mini-batch $\mathcal{B}$, let $(U^{(T)}, V^{(T)})$ denote the teacher's (frozen) text and image representations on $\mathcal{B}$, and let $(U_t^{(S)}, V_t^{(S)})$ and $(U_{t+1}^{(S)}, V_{t+1}^{(S)})$ denote the student's text and image representations on the *same* mini-batch $\mathcal{B}$ at optimization steps $t$ and $t+1$, respectively.

**Transfer entropy across optimization steps.** We define the teacher→student TE on $\mathcal{B}$ at step $t$ as the conditional mutual information

$$T_{\mathrm{T}\to\mathrm{S}}(t; \mathcal{B}) = I\Big( \big( U_{t+1}^{(S)}, V_{t+1}^{(S)} \big) ; \big( U^{(T)}, V^{(T)} \big) \,\Big|\, \big( U_t^{(S)}, V_t^{(S)} \big) \Big), \tag{3}$$

where $I(\cdot; \cdot \mid \cdot)$ is conditional mutual information. Intuitively, $T_{\mathrm{T}\to\mathrm{S}}(t; \mathcal{B})$ captures how much *additional* predictive information the teacher provides about the student's updated representations at step $t+1$, beyond what is already contained in the student's current representations at step $t$.

A high TE indicates that the teacher is meaningfully shaping the student's refinement from $(U_t^{(S)}, V_t^{(S)})$ to $(U_{t+1}^{(S)}, V_{t+1}^{(S)})$. This is especially expected early in distillation, when the student representations are underdeveloped and the teacher's signals provide substantial guidance. Tracking TE over training therefore provides a principled diagnostic of whether and when the student continues to benefit from the teacher, and whether the two modalities receive balanced guidance.

**Clarification on "time" and practicality.** One might worry that (3) assumes temporal correlation within a shuffled mini-batch. We emphasize that the index $t$ in (3) is not wall-clock time and does not refer to temporally correlated samples (e.g., videos). Instead, $t$ denotes the student's optimization step and (3) serves as a conceptual information-flow diagnostic.

At CLIP-scale dimensions, directly estimating (3) is intractable because it would require high-dimensional density estimation over cross-step joint variables. In practice, we therefore introduce the TE proxies in Section 4.2, which are tractable geometric surrogates motivated by Theorem 1. The proxy computation does not require any additional teacher evaluations beyond standard distillation: the teacher forward is computed once per mini-batch (without gradients) and reused to form within-batch finite-difference directions. TE then adds only lightweight difference and cosine-alignment computations on the current batch embeddings (no cache-and-replay and no extra forward/backward passes). Such information-theoretic perspectives on learning dynamics are consistent with prior work (Goldfeld et al., 2019; Achille & Soatto, 2018).

## 4.2 Approximating TE Using Cosine Similarity

In VLM distillation, computing *exact* transfer entropy is challenging because image and text representations are high-dimensional. For example, CLIP ResNet-50 produces 1024-dimensional embeddings (Radford et al., 2021). Since transfer entropy can be written as conditional mutual information (Eq. 2), evaluating $T_{X\to Y}$ requires estimating the joint density $p(y_{t+1}, y_t, x_t)$ and the conditionals $p(y_{t+1} \mid y_t, x_t)$ and $p(y_{t+1} \mid y_t)$. When $x_t$ and $y_t$ are continuous representations with $D \approx 10^3$, this becomes a high-dimensional density-estimation problem: even coarse discretization scales exponentially with dimension (the curse of dimensionality) (Köppen, 2000), and common TE/CMI estimators either rely on local neighborhood statistics (whose sample complexity grows rapidly in $D$) or impose parametric assumptions that are difficult to justify in deep feature spaces (Shahsavari Baboukani et al., 2020; Gowri et al., 2025). Moreover, TE depends on *joint* dependencies among $(y_{t+1}, y_t, x_t)$; capturing these interactions often entails estimating dependency structures with $\Theta(D^2)$ parameters (e.g., covariances or precision matrices) and expensive linear-algebra operations, or performing high-dimensional neighbor searches that become unreliable as $D$ increases. Collectively, these factors make exact TE estimation prohibitive for CLIP-scale embeddings, motivating the efficient TE surrogates we introduce next. We therefore propose the following theorem as the theoretical basis for our TE approximation.

**Theorem 1** (First-order TE–Jacobian relation)**.** *Let $x \in \mathbb{R}^d$ be an input (image–caption pair), and let $f_T, f_S : \mathbb{R}^d \to \mathbb{R}^D$ denote the teacher and student encoders with Jacobians*

$$J_T(x) = \nabla_x f_T(x), \quad J_S(x) = \nabla_x f_S(x) \in \mathbb{R}^{D \times d}. \tag{4}$$

*Under a first-order linear–Gaussian approximation of the conditional mutual information, the one-step transfer entropy from the teacher to the student satisfies*

$$T_T^S(x) \propto \cos\big(\widetilde{J}_S(x), \, \widetilde{J}_T(x)\big), \tag{5}$$

*where the Frobenius-normalized Jacobians are*

$$\widetilde{J}_S(x) = \frac{J_S(x)}{\|J_S(x)\|_F}, \qquad \widetilde{J}_T(x) = \frac{J_T(x)}{\|J_T(x)\|_F}, \tag{6}$$

*and $\cos(A, B) = \langle A, B \rangle_F$ denotes the cosine similarity (Frobenius inner product) between matrices $A$ and $B$.*

In Appendix C, we provide the proof for this theorem.

In practice, we approximate the Jacobians using finite differences (Nocedal & Wright, 1999)(Baydin et al., 2018):

$$J_S \delta x \rightsquigarrow f_S(x + \delta x) - f_S(x), \qquad J_T \delta x \rightsquigarrow f_T(x + \delta x) - f_T(x),$$

where $\delta x$ is a small input perturbation. Based on the these theoretical results, we propose two approximations on TE using cosine similarity.

**Ideal TE vs. our TE proxy.** Equation (3) defines an ideal teacher→student transfer-entropy quantity across optimization steps: it measures how much information the teacher's representation $(U^{(T)}, V^{(T)})$ transfers to the student's next-step state $(U_{t+1}^{(S)}, V_{t+1}^{(S)})$, beyond what is already explained by the student's current state $(U_t^{(S)}, V_t^{(S)})$. At CLIP-scale dimensions, directly estimating this conditional mutual information is intractable, as it would require high-dimensional density estimation over cross-step joint variables. Accordingly, Eq. (3) should be understood as the conceptual target (the teacher drives the student update), while our method uses a tractable geometric proxy motivated by first-order local approximation. In particular, Eq. (5) characterizes a perturbation-defined one-step teacher→student TE and motivates a cosine-based proxy via Jacobian alignment (proved in Appendix C). In our implementation (Algorithm 1), we instantiate this proxy using input-induced finite differences within the current mini-batch for both teacher and student.

**When is this alignment meaningful?** The TE proxies reward teacher-consistent geometry in representation space: for paired examples within a batch, they encourage the student's directional changes between examples to align with the corresponding teacher directional changes. This provides a relational learning signal that complements standard matching losses by transferring local structure of the teacher embedding space to the student. Throughout, we refer to Eq. (5) and TE1/TE2 as TE proxies rather than the exact optimization-step TE of Eq. (3).

**From Theorem 1 to practical TE proxies.** Theorem 1 provides a geometric characterization of a perturbation-defined one-step teacher→student transfer entropy, where the state transition is induced by a small input perturbation $x \mapsto x + \delta x$ (Appendix C). Under a first-order linear–Gaussian approximation, this quantity satisfies $T_T^S(x) \propto \cos(\widetilde{J}_S(x), \widetilde{J}_T(x))$. In contrast, Eq. (3) defines an ideal TE across optimization steps (parameter updates), which is generally intractable to estimate directly in high-dimensional feature spaces. We therefore use the Jacobian-alignment signal from Theorem 1 as an efficient TE proxy without claiming to directly estimate the optimization-step TE in Eq. (3).

In practice, we avoid explicit Jacobian construction by using finite differences on representations. Within a mini-batch $\mathcal{B} = \{x_i\}_{i=1}^B$, the index $i$ enumerates examples and is not treated as time; thus our proxy should not be interpreted as estimating temporal TE from shuffled batches. Instead, we form finite-difference directional changes across paired inputs within the same mini-batch: $\Delta f_T(x_i) := f_T(x_{i'}) - f_T(x_i)$ and

$\Delta f_S(x_i) := f_S(x_{i'}) - f_S(x_i)$ for paired examples $(x_i, x_{i'})$. Under local linearization, these differences serve as practical surrogates for Jacobian actions, $\Delta f(x) \approx J(x)(x_{i'} - x_i)$. Cosine similarity between teacher and student differences therefore provides a tractable estimator of the Jacobian-alignment term suggested by Theorem 1.

We refer to the resulting cosine-based objectives as TE1 (Section 4.2.1) and TE2 (Section 4.2.2), and treat them as geometric TE proxies motivated by Theorem 1 and implemented in Algorithm 1.

### 4.2.1 TE Approximation via Cosine Similarity of Differences

Let $\mathbf{v}^{(S)}$ and $\mathbf{u}^{(S)}$ denote the image and text embeddings from the student model, and $\mathbf{v}^{(T)}$ and $\mathbf{u}^{(T)}$ denote the corresponding embeddings from the teacher model. The TE approximations are based on computing the cosine similarity between the local *variation across examples* in the student and teacher embedding spaces on the same mini-batch. Since the teacher is frozen, any variation in teacher embeddings arises from changing inputs, not from training-time dynamics.

To approximate TE, the method first calculates the difference between consecutive embeddings for both image and text modalities. The embedding differences are computed as

$$\Delta \mathbf{v}_i^{(S)} = \mathbf{v}_{i+1}^{(S)} - \mathbf{v}_i^{(S)}, \quad \Delta \mathbf{v}_i^{(T)} = \mathbf{v}_{i+1}^{(T)} - \mathbf{v}_i^{(T)} \tag{7}$$

for images, and

$$\Delta \mathbf{u}_i^{(S)} = \mathbf{u}_{i+1}^{(S)} - \mathbf{u}_i^{(S)}, \quad \Delta \mathbf{u}_i^{(T)} = \mathbf{u}_{i+1}^{(T)} - \mathbf{u}_i^{(T)} \tag{8}$$

for text embeddings.

Here, the index $i$ enumerates examples within a mini-batch and does not represent time. We form differences between pairs of examples in the *same* batch (e.g., adjacent elements after an internal random permutation) and use TE1/TE2 as *geometric* finite-difference surrogates, not as temporal TE estimators from shuffled data. Thus, TE1 and TE2 are geometric surrogates that encourage the student to match the teacher's local embedding-space geometry, and do not assume temporal structure in the data ordering.

Once the differences are obtained, the next step involves computing the cosine similarity between the student's and teacher's directional changes. Cosine similarity (Xia et al., 2015) serves as a measure of alignment between the two models, ensuring that if the student's representation updates closely follow the teacher's, meaningful information transfer is occurring. The cosine similarity for images is given by

$$\cos \theta_i^{(v)} = \frac{\langle \Delta \mathbf{v}_i^{(S)}, \Delta \mathbf{v}_i^{(T)} \rangle}{\|\Delta \mathbf{v}_i^{(S)}\| \|\Delta \mathbf{v}_i^{(T)}\| + \epsilon}, \tag{9}$$

where $\epsilon$ is a small constant to prevent division by zero. While for text embeddings, it is given by

$$\cos \theta_i^{(u)} = \frac{\langle \Delta \mathbf{u}_i^{(S)}, \Delta \mathbf{u}_i^{(T)} \rangle}{\|\Delta \mathbf{u}_i^{(S)}\| \|\Delta \mathbf{u}_i^{(T)}\| + \epsilon}. \tag{10}$$

To approximate the overall TE for each modality, the method computes the mean cosine similarity across all batch elements. The image-based TE is computed as

$$\text{TE}_{\text{img}} = \frac{1}{B-1} \sum_{i=1}^{B-1} \cos \theta_i^{(v)}, \tag{11}$$

while the text-based TE follows the same formulation:

$$\text{TE}_{\text{txt}} = \frac{1}{B-1} \sum_{i=1}^{B-1} \cos \theta_i^{(u)}. \tag{12}$$

The final TE approximation (TE1) is obtained by averaging the image and text TE values:

$$\text{TE1} = \frac{1}{2} (\text{TE}_{\text{img}} + \text{TE}_{\text{txt}}). \tag{13}$$

### 4.2.2 TE Approximation via Cosine Similarity of Concatenated Differences

An alternative approach to approximate TE involves combining the directional changes from both image and text modalities before computing the cosine similarity. In this method, we first calculate the differences between consecutive embeddings for both modalities, same as (7)(8). Instead of computing the cosine similarity for each modality independently and then averaging the results, we concatenate the difference vectors from both modalities into a single vector. That is, for each index $i$, we define the concatenated difference vectors as

$$\Delta \mathbf{c}_i^{(S)} \quad = \quad \left[ \Delta \mathbf{v}_i^{(S)} \, \| \, \Delta \mathbf{u}_i^{(S)} \right], \tag{14}$$

$$\Delta \mathbf{c}_i^{(T)} \quad = \quad \left[ \Delta \mathbf{v}_i^{(T)} \, \| \, \Delta \mathbf{u}_i^{(T)} \right], \tag{15}$$

where $\|$ denotes concatenation along the feature dimension.

The cosine similarity between the concatenated difference vectors is then computed as

$$\cos \theta_i^{(\mathrm{cat})} = \frac{\langle \Delta \mathbf{c}_i^{(S)}, \Delta \mathbf{c}_i^{(T)} \rangle}{\| \Delta \mathbf{c}_i^{(S)} \| \, \| \Delta \mathbf{c}_i^{(T)} \| + \epsilon}, \tag{16}$$

with $\epsilon$ being a small constant for numerical stability. Finally, the overall TE approximation (TE2) is obtained by averaging these cosine similarities over all consecutive pairs in the batch:

$$\mathrm{TE2} = \frac{1}{B-1} \sum_{i=1}^{B-1} \cos \theta_i^{(\mathrm{cat})}. \tag{17}$$

This concatenation-based surrogate for TE captures the joint evolution of image and text representations, providing a single metric that reflects how well the student model's combined modality updates align with those of the teacher.

**Why concatenation is meaningful (and not equivalent to averaging).** We emphasize that TE2 is *not* intended to be algebraically equivalent to averaging the per-modality cosine similarities (TE1), since cosine similarity is nonlinear. Rather, TE2 provides a *single joint geometric score* by measuring the angle between the *concatenated* update vectors $\Delta \mathbf{c}_i^{(S)} = [\Delta \mathbf{v}_i^{(S)} \, \| \, \Delta \mathbf{u}_i^{(S)}]$ and $\Delta \mathbf{c}_i^{(T)} = [\Delta \mathbf{v}_i^{(T)} \, \| \, \Delta \mathbf{u}_i^{(T)}]$. Geometrically, this is equivalent to computing cosine similarity in the direct-sum space $\mathbb{R}^{d_v} \oplus \mathbb{R}^{d_u}$, i.e., enforcing *joint* alignment of the image and text update directions. By contrast, TE1 treats the two modalities independently and averages their alignment scores.

Writing $a = \Delta \mathbf{v}_i^{(S)}$, $b = \Delta \mathbf{u}_i^{(S)}$, $c = \Delta \mathbf{v}_i^{(T)}$, $d = \Delta \mathbf{u}_i^{(T)}$, we can expand TE2 as

$$\cos \theta_i^{(\mathrm{cat})} = \frac{\langle a, c \rangle + \langle b, d \rangle}{\sqrt{\|a\|^2 + \|b\|^2} \, \sqrt{\|c\|^2 + \|d\|^2} + \epsilon}. \tag{18}$$

Equation (18) shows that TE2 couples the modalities through the *shared normalization*: a modality with large update magnitude (e.g., image) will dominate the joint direction unless the other modality (e.g., text) is also aligned. This coupling is desirable in VLM distillation because it encourages the student to align the *combined* multimodal update geometry to the teacher, rather than achieving high alignment in only one modality.

In Appendix D, we present evaluation results for our two TE approximation methods and compare them with exact TE computation in simple experimental settings. In Appendix E, we analyze the computational cost of exact TE versus TE approximations. In all experiments, TE and its surrogates are computed on the *global* CLIP embeddings. For images, $V$ denotes the pooled visual embedding (e.g., global representation / CLS token for ViT). For text, $U$ denotes the embedding at the end-of-text (EOT) token position in the final Transformer layer. We do not operate on token-level features; all TE quantities are defined on these pooled representations.

### 4.3 Loss Functions for VLM Distillation

**Motivating the combined objective.** Our total objective combines complementary distillation signals that operate at different granularities. The *contrastive loss* (CL) preserves CLIP-style global alignment by learning a discriminative cross-modal embedding space within a batch (matching image-text pairs while separating negatives). The *KL divergence* term (KL) further distills the teacher's *soft similarity distribution* over the batch, transferring relative-neighbor structure ("dark knowledge") beyond hard positives. The *feature regression* term (MSE/L2) anchors the student to the teacher in representation space to stabilize training and reduce collapse/drift, especially when the student is lower-capacity. *Interactive contrastive learning* (ICL) explicitly enforces cross-modal consistency between student and teacher (student image vs. teacher text and student text vs. teacher image), which improves modality coupling and helps the student inherit the teacher's cross-modal correspondences.

**What TE contributes beyond CL/KL/MSE/ICL.** CL/KL/MSE/ICL mainly encourage *static matching* of embeddings and similarity structure at a given step. In contrast, the TE term targets a different signal: it encourages the teacher to *drive the student's update direction* by rewarding alignment between the student's representation change and the teacher-implied change (our efficient geometric surrogate for TE). Empirically, we find TE complements the above losses by improving the *update dynamics* (i.e., whether the student moves in teacher-consistent directions) rather than only reducing instantaneous discrepancies, which yields more robust retrieval gains under capacity gaps and training noise.

To effectively transfer knowledge from the teacher model to the student model in a VLM distillation setting, we employ a weighted combination of CL, KL divergence, MSE feature distillation, ICL, a mutual-information (MI) loss (Tian et al., 2020), an MSE-diff (MSE-$\Delta$) loss, and a TE surrogate reward. All loss terms (except TE) are defined in Appendix F. We include MI and MSE-diff as established geometric baselines for fair comparison and ablation.

To integrate the transfer entropy component, we treat TE as a reward and subtract its surrogate score from the overall loss. Combining all terms, the total loss function becomes:

$$\mathcal{L}_{\text{total}} = \mathcal{L}_{\text{contrastive}} + \alpha \, \mathcal{L}_{\text{KL}} + \beta \, \mathcal{L}_{\text{MSE}} + \delta \, \mathcal{L}_{\text{ICL}} + \eta \, \mathcal{L}_{\text{MI}} + \lambda \, \mathcal{L}_{\text{MSE-}\Delta} - \gamma \, \text{TE}, \tag{19}$$

where $\alpha$, $\beta$, $\delta$, $\eta$, $\lambda$, and $\gamma$ are weighting factors that balance the contributions of the KL divergence loss, the MSE loss, the ICL loss, the MI loss, the MSE-$\Delta$ loss, and the TE reward, respectively. Here, $\mathcal{L}_{\text{MI}}$ is implemented as an InfoNCE-style contrastive objective that lower-bounds mutual information. Minimizing $\mathcal{L}_{\text{MI}}$ encourages matched teacher-student representations to score higher than mismatched pairs within the batch (Tian et al., 2020).

This composite loss encourages the student model to align with the teacher's similarity distributions and feature space while also capturing change-based signals in representation evolution, resulting in more faithful distillation of multi-modal interactions. By integrating CL, KL divergence, MSE loss, ICL loss, MI and MSE-$\Delta$ losses, and TE-based regularization, we construct a comprehensive loss function that balances distributional alignment, feature matching, and change-aware supervision for effective VLM distillation.

### 4.4 Description of a Full Training Step with TE Approximation

**One training step with TE surrogates (within-batch finite differences).** Algorithm 1 summarizes a full distillation iteration and makes explicit how our TE surrogates (TE1/TE2) are computed in practice. At each iteration, we sample one mini-batch $\mathcal{B} = \{(x_i, y_i)\}_{i=1}^{B}$ and perform *one* student parameter update. The teacher $f_T$ is frozen; therefore, its embeddings provide a stable reference geometry for the current batch, while the student $f_S$ is updated by backpropagation through the overall objective.

Crucially, TE1/TE2 do *not* require any extra teacher evaluation beyond standard distillation. We run the teacher forward once (without gradients) on the current batch and detach the resulting embeddings $(\mathbf{v}^{(T)}, \mathbf{u}^{(T)})$. We also run the student forward once on the *same* batch to obtain $(\mathbf{v}^{(S)}, \mathbf{u}^{(S)})$. To form a finite-difference proxy for local directional structure, we first sample a random permutation $\pi$ of the batch indices and reorder the batch as $\mathcal{B}_\pi$. We then construct *input-induced* within-batch differences using adjacent

---

**Algorithm 1:** One distillation step with TE1/TE2 (within-batch finite differences)

---

**Input:** Student $f_S$, frozen teacher $f_T$, batch $\mathcal{B} = \{(x_i, y_i)\}_{i=1}^B$

**Output:** Updated student parameters $\theta_S$

Sample mini-batch $\mathcal{B}$;

Sample a random permutation $\pi$ over $\{1, \ldots, B\}$;

Form permuted batch $\mathcal{B}_\pi = \{(x_{\pi(i)}, y_{\pi(i)})\}_{i=1}^B$;

**Teacher (once, no grad):** $(\mathbf{v}^{(T)}, \mathbf{u}^{(T)}) \leftarrow f_T(\mathcal{B}_\pi)$;

Detach teacher features;

**Student forward:** $(\mathbf{v}^{(S)}, \mathbf{u}^{(S)}) \leftarrow f_S(\mathcal{B}_\pi)$;

Compute base distillation loss $\mathcal{L}_{\text{base}} \leftarrow \mathcal{L}_{\text{CL}} + \alpha\mathcal{L}_{\text{KL}} + \beta\mathcal{L}_{\text{L2}} + \delta\mathcal{L}_{\text{ICL}} + \eta\,\mathcal{L}_{\text{MI}} + \lambda\,\mathcal{L}_{\text{MSE-}\Delta}$;

`// Within-batch finite differences (adjacent pairs after permutation)`

**for** $i = 1$ **to** $B - 1$ **do**

    $\Delta\mathbf{v}_i^{(S)} \leftarrow \mathbf{v}_{i+1}^{(S)} - \mathbf{v}_i^{(S)}$,;

    $\Delta\mathbf{u}_i^{(S)} \leftarrow \mathbf{u}_{i+1}^{(S)} - \mathbf{u}_i^{(S)}$;

    $\Delta\mathbf{v}_i^{(T)} \leftarrow \mathbf{v}_{i+1}^{(T)} - \mathbf{v}_i^{(T)}$,;

    $\Delta\mathbf{u}_i^{(T)} \leftarrow \mathbf{u}_{i+1}^{(T)} - \mathbf{u}_i^{(T)}$;

`// TE1:  per-modality alignment`

$\text{TE}_{\text{img}} \leftarrow \frac{1}{B-1} \sum_{i=1}^{B-1} \cos(\Delta\mathbf{v}_i^{(S)}, \Delta\mathbf{v}_i^{(T)})$;

$\text{TE}_{\text{txt}} \leftarrow \frac{1}{B-1} \sum_{i=1}^{B-1} \cos(\Delta\mathbf{u}_i^{(S)}, \Delta\mathbf{u}_i^{(T)})$;

$\text{TE1} \leftarrow \frac{1}{2}(\text{TE}_{\text{img}} + \text{TE}_{\text{txt}})$;

`// TE2:  joint multimodal alignment`

**for** $i = 1$ **to** $B - 1$ **do**

    $\Delta\mathbf{c}_i^{(S)} \leftarrow [\Delta\mathbf{v}_i^{(S)} \,\|\, \Delta\mathbf{u}_i^{(S)}]$;

    $\Delta\mathbf{c}_i^{(T)} \leftarrow [\Delta\mathbf{v}_i^{(T)} \,\|\, \Delta\mathbf{u}_i^{(T)}]$;

$\text{TE2} \leftarrow \frac{1}{B-1} \sum_{i=1}^{B-1} \cos(\Delta\mathbf{c}_i^{(S)}, \Delta\mathbf{c}_i^{(T)})$;

Set total loss: $\mathcal{L} \leftarrow \mathcal{L}_{\text{base}} - \gamma(\text{TE1} + \text{TE2})$;

Update student parameters: $\theta_S \leftarrow \theta_S - \eta\nabla_{\theta_S}\mathcal{L}$;

---

pairs in the permuted order: $\Delta\mathbf{v}_i = \mathbf{v}_{i+1} - \mathbf{v}_i$ and $\Delta\mathbf{u}_i = \mathbf{u}_{i+1} - \mathbf{u}_i$ (for both teacher and student). These differences should be interpreted as *geometric* directions between nearby examples in embedding space. The index $i$ enumerates permuted samples and carries *no* temporal semantics; thus TE1/TE2 are not temporal TE estimators from shuffled data. Instead, TE1 rewards per-modality cosine alignment between teacher and student directions, while TE2 measures joint multimodal alignment by concatenating image and text directions before computing cosine similarity. Together, these rewards encourage the student to match the teacher's local representation geometry on each batch in a relational, direction-field sense.

**Compute footprint.** Per iteration, we run the teacher forward once (no gradients) and the student forward once (with gradients), as in standard distillation. TE1/TE2 add only lightweight vector differences and cosine computations on the already-computed embeddings, and therefore incur negligible overhead (no additional forward passes and no cache-and-replay).

## 5 Experiments

Our experiments consist of the following configurations: (1) Teacher: ResNet-50, Student: ResNet-34; (2) Teacher: ViT-B/16, Student: ResNet-34; (3) Teacher: ResNet-50×16, Student: ResNet-34; (3) Teacher: ResNet-50, Student: ResNet-18. The introduction of CLIP RN50, Vit-B/16, and RN50×16 are introduced in Appendix G. We evaluate these settings on three datasets: MSCOCO 2014 (Lin et al., 2014), Flickr8k (Hodosh et al., 2013)(Marco et al., 2023), and Flick30k (Young et al., 2014). We also include one application

in classification based on Food 101 dataset (Bossard et al., 2014) and zero-shot evaluated of distilled RN34 using ImageNet-1k (Deng et al., 2009)(Russakovsky et al., 2015) in Appendix H.3 and H.4.

## 5.1 Teacher: RN50, Student: RN34, Dataset: MSCOCO

The student VLM model is based on RN34 for the image encoder and a lightweight Transformer for the text encoder. The RN34 architecture contains approximately 21.8 million parameters, and the final fully connected layer is modified to output 1024-dimensional features, keeping the parameter count relatively stable (He et al., 2016). The text encoder consists of an embedding layer with a vocabulary size of 49,408 and a hidden dimension of 1024, contributing approximately 25.3 million parameters (Mehta et al., 2020). Additionally, the student Transformer has only 2 encoder layers with an 8-head attention mechanism, leading to an estimated total of 5-10 million parameters (Vaswani et al., 2017). Combining both encoders, the total parameter count of the student model is approximately 55-60 million, significantly smaller than the teacher model while maintaining effective knowledge representation capabilities.

Our experiments are conducted on the MSCOCO 2014 dataset (Lin et al., 2014), which comprises approximately 82,783 training images and 40,504 validation images, each paired with multiple textual descriptions. This dataset is widely adopted in vision-language research due to its extensive and diverse image-caption pairs.

The student model is trained using various combinations of loss functions, including Contrastive Loss (CL), KL divergence, MSE loss, ICL loss, and our proposed TE rewards. The total loss function is defined in (19). We conducted experiments using different combinations of these loss components. Our TE-based regularization is designed to capture the directional information flow between the teacher and student feature encoders, thereby encouraging the student to mimic the teacher's behavior more closely.

The hyperparameters $\alpha$, $\beta$, $\delta$, and $\gamma$ in (19) are designed to balance the contributions of the CL, KL, MSE, ICL, and TE terms in the overall objective. We assign larger values to hyperparameters associated with loss components that naturally exhibit smaller magnitudes, ensuring that each term contributes comparably to the optimization process. The training losses and TE for different loss functions are provided in Fig. 1.

In this experiment, we used weighting factors $\alpha = 1.0$, $\beta = 50$, $\delta = 1.0$, $\gamma = 1.0$, and a temperature parameter $\tau = 0.07$. These hyperparameters were carefully selected to balance the contributions of each loss component, ensuring effective knowledge transfer from the teacher to the student model while maintaining training stability. The batch size was set to $|B| = 64$, and the training data was shuffled to eliminate correlations between neighboring samples.

We utilized Google Colab Pro with a T4 GPU and High-RAM for training and performance evaluation. Due to time and budget constraints, we trained the student model, RN34, for only 10 epochs in each loss function combination scenario. The training and evaluation process for each experimental setup took approximately 14 hours.

Figure 1 illustrates that the total training loss decreases steadily over epochs while the TE rewards show an increasing trend. This behavior indicates that the model effectively minimizes the overall objective and progressively captures the directional information flow between teacher and student representations. The TE-based regularization plays a key role in maintaining structured alignment during training, which is critical for effective knowledge transfer. For experiments with TE1 and TE2 such as Fig. 1h and Fig. 1i, the TE1 and TE2 monotonically increase with very close but different values. However, the KL loss and MSE don't decrease clearly with more training epochs. In loss functions with KL, ICL, MSE, TE, different combinations may impact each other. For example, in Contrastive - TE1 (Fig. 1d), TE1 achieved average value 0.7242 at epoch 10; in Contrastive + KL - TE1 (Fig. 1a), TE1 achieved average value 0.7865 at epoch 10; and in Contrastive + KL + ICL - TE1 (Fig. 1e), TE1 achieved average value 0.4611 at epoch 10. So this demonstrated that KL promotes TE, but ICL discourages TE.

We evaluated the performance of the trained student models using Recall@k for both image-to-text (I2T) and text-to-image (T2I) retrieval tasks. Recall@k measures the percentage of queries for which the correct match appears in the top-k retrieved results (Manning et al., 2008). A higher Recall@1 indicates stronger alignment between images and texts, as the correct match is ranked first, while Recall@5 and Recall@10

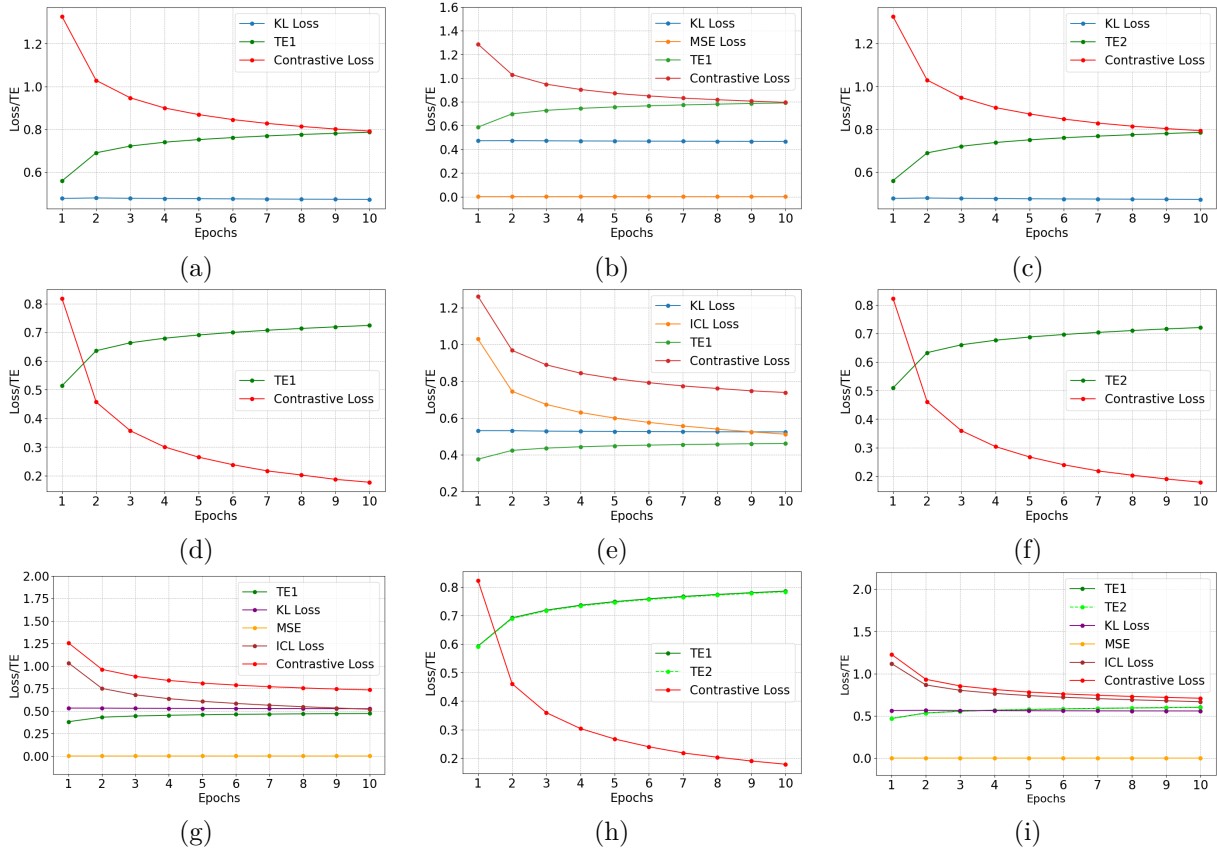

Figure 1: The training losses and TE for different loss functions in the distillation training of RN34-based VLM student with teacher CLIP RN50 using MSCOCO dataset. (a) Contrastive + KL - TE1, (b) Contrastive + KL + MSE - TE2, (c) Contrastive +KL - TE2, (d) Contrastive - TE1, (e) Contrastive + KL + ICL - TE1, (f) Contrastive - TE2, (g) Contrastive + KL + +MSE + ICL - TE1, (h) Contrastive - TE1 - TE2, (i) Contrastive + KL + ICL + MSE -TE1 - TE2.

provide insight into broader retrieval accuracy. We summarized the performance of the evaluation of the trained student model RN34 in Table 1.

Table 1 clearly demonstrates that incorporating transfer entropy (TE1 and TE2) into the VLM distillation objective leads to substantial performance gains in both I2T and T2I retrieval. We incrementally added each loss component to the base contrastive loss and observed that the introduction of TE1 or TE2 resulted in the most significant performance improvements. The best-performing models all include TE components, underscoring their effectiveness in enhancing the student model's ability to capture structured information flow from the teacher. Notably, the configuration using the loss function `CL - TE1 - TE2` achieves the highest I2T Recall@1, while the full loss combination `CL + KL + MSE + ICL - TE1 - TE2` yields the best T2I performance. This suggests that TE terms not only provide strong standalone regularization but also complement traditional distillation objectives when integrated holistically. As shown in Figure 1f, both TE1 and TE2 exhibit similar trends with monotonically increasing values during training. This indicates that their influence becomes more prominent over time, effectively guiding the optimization of the CL–TE1–TE2 objective. We highlight the best-performing scores in bold in Table 1.

Table 2 reports the sensitivity of retrieval performance to different hyperparameter settings. Overall, the method is reasonably stable to moderate changes in loss weights, while the TE reward weight $\gamma$ produces the most consistent and largest gains. Starting from the baseline without TE ($\gamma = 0$), introducing a small TE weight improves retrieval in both directions, and performance generally increases as $\gamma$ grows from 1 to 7.5. In particular, $\gamma = 7.5$ achieves the best I2T results (R@1/5/10 of 10.27/26.36/36.30),

Table 1: Comparison of zero-shot retrieval performance (Recall@k) of RN34-based VLM student with teacher CLIP RN50 for different loss function combinations in VLM distillation using **MSCOCO**. All Loss Function: CL + KL + MSE + ICL - TE1 - TE2. The weighting factors for loss functions: $\alpha = 1.0$, $\beta = 50$, $\delta = 1.0$, $\gamma = 1.0$.

| Model and Loss Function | I2T Retrieval (R) | | | T2I Retrieval (R) | | |
|---|---|---|---|---|---|---|
| | R@1 | R@5 | R@10 | R@1 | R@5 | R@10 |
| **Teacher Model (RN50)** | 15.27% | 30.73% | 39.05% | 11.68% | 25.52% | 33.50% |
| **Student Models (RN34)** | | | | | | |
| CL Only (Oord et al., 2018) | 4.94% | 14.60% | 22.51% | 3.96% | 12.67% | 19.45% |
| CL + MSE (Yang et al., 2024) | 5.13% | 15.41% | 23.17% | 4.00% | 12.79% | 19.53% |
| CL + KL (Li et al., 2024b) | 5.42% | 16.20% | 24.55% | 5.06% | 15.35% | 22.92% |
| CL + ICL (Yang et al., 2024) | 5.75% | 16.86% | 24.83% | 5.07% | 15.14% | 22.44% |
| CL - TE1 | 6.91% | 19.48% | 28.12% | 5.68% | 16.33% | 23.93% |
| CL - TE2 | 7.04% | 19.22% | 27.97% | 5.46% | 15.90% | 23.49% |
| CL - TE1 - TE2 | **8.24**% | **22.43**% | **31.73**% | 6.53% | 18.13% | 26.02% |
| CL + KL - TE1 | 7.81% | 21.31% | 30.65% | 6.42% | 18.18% | 26.33% |
| CL + KL - TE2 | 7.77% | 21.10% | 30.22% | 6.21% | 17.95% | 26.03% |
| CL + KL + MSE - TE1 | 7.62% | 21.05% | 30.34% | 6.53% | 18.48% | 26.66% |
| CL + KL + ICL - TE1 | 7.59% | 20.87% | 29.95% | 6.78% | 19.11% | 27.33% |
| CL + KL + MSE + ICL - TE1 | 7.51% | 20.62% | 29.81% | 6.76% | 19.02% | 27.32% |
| All Loss Function | 8.11% | 22.05% | 31.57% | **7.18**% | **19.75**% | **28.14**% |

indicating that TE regularization meaningfully strengthens cross-modal alignment when used as a reward term. Increasing $\gamma$ further (e.g., $\gamma = 10$) yields diminishing returns and slightly degrades performance, suggesting that overly strong TE weighting can over-regularize the student. By comparison, varying $\alpha$ (KL), $\beta$ (MSE), and $\delta$ (ICL) within the tested ranges leads to smaller fluctuations, implying that the objective is not overly sensitive to these terms once they are set to reasonable values. Adding the MI loss also improves retrieval, especially for T2I: setting $\eta = 2.5$ (with $\gamma = 0$) yields the strongest T2I performance in Table 2 (R@1/5/10 of 7.66/20.69/29.25), supporting the role of MI-based contrastive supervision in improving bi-directional correspondence. In contrast, using the MSE-$\Delta$ loss alone ($\lambda \in \{50, 100\}$ with $\gamma = 0$) provides only marginal gains over the baseline, and combining multiple auxiliary terms does not always yield additive improvements, highlighting the importance of balancing regularizers. Based on these trends, we adopt moderate TE weighting (around $\gamma = 7.5$) and MI weighting (around $\eta = 2.5$) as effective default settings for the remaining experiments.

**Hyperparameter selection and reproducibility.** The weights $\alpha, \beta, \delta, \eta, \lambda$, and $\gamma$ balance the relative influence of the KL, MSE (feature regression), ICL, MI, MSE-$\Delta$, and TE surrogate terms against the base contrastive objective. By "assigning larger weights to losses with smaller magnitudes," we mean the following concrete heuristic: during a short warm-up run (first ~1 epoch) with a fixed reference setting, we log the *batch-averaged* value of each unweighted component $\{\mathcal{L}_{\text{CL}}, \mathcal{L}_{\text{KL}}, \mathcal{L}_{\text{MSE}}, \mathcal{L}_{\text{ICL}}, \mathcal{L}_{\text{MI}}, \mathcal{L}_{\text{MSE}\Delta}, \text{TE}_1, \text{TE}_2\}$, and choose weights so that the *weighted* components have comparable scale:

$$\alpha \, \mathbb{E}[\mathcal{L}_{\text{KL}}] \approx \beta \, \mathbb{E}[\mathcal{L}_{\text{MSE}}] \approx \delta \, \mathbb{E}[\mathcal{L}_{\text{ICL}}] \approx \eta \, \mathbb{E}[\mathcal{L}_{\text{MI}}] \approx \lambda \, \mathbb{E}[\mathcal{L}_{\text{MSE}\Delta}] \approx \gamma \, \mathbb{E}[\text{TE}] \approx \mathbb{E}[\mathcal{L}_{\text{CL}}], \tag{20}$$

where TE denotes the (bounded) cosine-based surrogate (TE$_1$/TE$_2$). Intuitively, when a term is naturally small (e.g., MSE between normalized embeddings), a larger coefficient is required for it to have a non-negligible effect on the update; conversely, terms already on the same order as $\mathcal{L}_{\text{CL}}$ require smaller coefficients.

After setting a sensible scale using (20), we perform a small grid search around these values and select the final configuration by validation Recall@K on MSCOCO. Table 2 reports this sweep and shows that performance is stable across a range of values; in our main experiments we fix $\alpha = 1$ and $\delta = 1$, use $\beta \in \{50, 100\}$ depending on the teacher backbone, and tune $\gamma$ from $\{1, 2.5, 5, 7.5, 10\}$ with $\gamma = 7.5$ working

Table 2: Comparison of zero-shot retrieval performance (Recall@k) in percentage of RN34-based VLM student with teacher CLIP RN50 on MSCOCO.

| $\alpha$ | $\beta$ | $\delta$ | $\eta$ | $\lambda$ | $\gamma$ | I2T R@1 | I2T R@5 | I2T R@10 | T2I R@1 | T2I R@5 | T2I R@10 |
|---|---|---|---|---|---|---|---|---|---|---|---|
| 1 | 50 | 1 | 0 | 0 | 0 | 6.08% | 18.14% | 26.93% | 5.92% | 17.06% | 24.89% |
| 1 | 50 | 1 | 0 | 0 | 1 | 8.11% | 22.05% | 31.57% | 7.18% | 19.75% | 28.14% |
| 1 | 50 | 1 | 0 | 0 | 2.5 | 9.48% | 24.68% | 34.54% | 7.57% | 20.55% | 29.03% |
| 1 | 50 | 1 | 0 | 0 | 5 | 10.02% | 25.80% | 35.74% | 7.32% | 20.04% | 28.50% |
| 1 | 50 | 1 | 0 | 0 | 7.5 | **10.27%** | **26.36%** | **36.30%** | 6.98% | 18.95% | 26.95% |
| 1 | 50 | 1 | 0 | 0 | 10 | 9.77% | 25.25% | 35.04% | 6.94% | 18.78% | 26.67% |
| 5 | 50 | 1 | 0 | 0 | 5 | 10.19% | 26.15% | 36.09% | 6.81% | 18.84% | 26.88% |
| 1 | 100 | 1 | 0 | 0 | 5 | 10.25% | 25.76% | 35.69% | 7.30% | 20.55% | 29.03% |
| 1 | 50 | 5 | 0 | 0 | 5 | 8.20% | 22.14% | 31.63% | 7.14% | 19.61% | 28.05% |
| 1 | 50 | 1 | 0 | 50 | 0 | 6.67% | 18.48% | 27.30% | 6.01% | 17.27% | 25.24% |
| 1 | 50 | 1 | 0 | 100 | 0 | 6.52% | 18.90% | 27.57% | 6.29% | 17.71% | 25.67% |
| 1 | 50 | 1 | 1 | 1 | 0 | 8.35% | 22.62% | 31.70% | 7.43% | 20.23% | 28.87% |
| 1 | 50 | 1 | 2.5 | 0 | 0 | 8.56% | 23.01% | 32.41% | **7.66%** | **20.69%** | **29.25%** |
| 1 | 50 | 1 | 5 | 0 | 0 | 8.65% | 23.29% | 33.05% | 7.65% | 20.63% | 29.05% |
| 1 | 50 | 1 | 7.5 | 0 | 0 | 8.25% | 22.41% | 32.04% | 7.44% | 20.27% | 28.61% |
| 1 | 50 | 1 | 2.5 | 50 | 2.5 | 9.12% | 24.25% | 33.88% | 7.31% | 19.98% | 28.40% |
| 1 | 50 | 1 | 2.5 | 0 | 5 | 9.94% | 25.73% | 35.54% | 6.87% | 18.92% | 27.09% |
| 1 | 50 | 1 | 2.5 | 50 | 5 | 9.26% | 24.26% | 34.44% | 7.05% | 19.47% | 27.78% |
| 1 | 50 | 1 | 1 | 0 | 7.5 | 9.67% | 24.85% | 34.78% | 6.91% | 18.95% | 27.07% |
| 1 | 50 | 1 | 1 | 50 | 7.5 | 10.23% | 25.93% | 35.81% | 6.87% | 18.91% | 27.04% |
| 1 | 50 | 1 | 5 | 50 | 7.5 | 9.03% | 24.15% | 33.87% | 6.85% | 18.80% | 26.77% |

best for RN50→RN34 on MSCOCO. For MI baselines we tune $\eta$ analogously (e.g., $\eta \in \{1, 2.5, 5, 7.5\}$), and we set $\lambda = 0$ unless explicitly evaluating the MSE-$\Delta$ ablation.

## 5.2 Teacher: ViT-B/16, Student: ResNet-34

We distill an RN34-based VLM student from a CLIP ViT-B/16 teacher. For this experiment, the loss function incorporates weighting factors $\alpha = 1.0$, $\beta = 100$, $\delta = 1.0$, $\gamma = 5.0$, along with a temperature parameter $\tau = 0.07$. These weighting parameters were chosen based on the relative contribution of each loss term to the total loss during training. For the experiments on the MSCOCO dataset, due to the large scale of both the model and the dataset, we trained the student model for 6 epochs. Each experiment (i.e., each row in Table 3) required approximately 10 hours on a Google Colab T4 GPU with high-RAM. For the Flickr8k experiments, we used a Google Colab A100 GPU and trained for 10 epochs. Given the smaller dataset size, each experiment (i.e., each row in Table 5) took around 30 minutes to complete.

Our experiments (Tables 3 and 5) show that maximizing the information flow from teacher to student via TE delivers the single largest boost among all losses. Loss functions with TE leading to the 3–4 percentage point (pp) gains on MSCOCO and the 8–12pp gains on the low-resource Flickr8k benchmarks. These results establish TE as a principled and highly effective regularizer for cross-modal knowledge distillation.

Table 4 summarizes the hyperparameter sensitivity of zero-shot retrieval when distilling a student RN34 from a ViT-B/16 teacher on MSCOCO. Overall, introducing the TE reward is consistently beneficial, especially for T2I retrieval: compared to the baseline without TE ($\gamma = 0$), a small TE weight ($\gamma = 1$) yields the strongest T2I scores in the table (R@1/5/10 of 6.75/18.74/26.91), indicating that TE-based regularization can effectively strengthen bidirectional alignment under this teacher-student pairing. Increasing $\gamma$ beyond this regime provides diminishing returns and can reduce performance (e.g., $\gamma \geq 5$), suggesting that overly strong TE weighting may over-regularize the student. We also observe that increasing the MSE weight $\beta$ can improve I2T retrieval: the best I2T performance is achieved at $\beta = 100$ with $\gamma = 5$ (R@1/5/10 of 8.48/22.02/30.99), highlighting the importance of feature-level matching for transferring visual semantics

Table 3: Comparison of zero-shot retrieval performance (Recall@k) of RN34-based VLM student with teacher CLIP ViT-B/16 on MSCOCO in VLM distillation with different loss functions. All Loss Function: CL + KL + MSE + ICL - TE1 - TE2. The hyperparameter settings: $\alpha = 1.0$, $\beta = 100$, $\delta = 1.0$, $\gamma = 5.0$.

| Model and Loss Function | I2T Retrieval (R) | | | T2I Retrieval (R) | | |
|---|---|---|---|---|---|---|
| | R@1 | R@5 | R@10 | R@1 | R@5 | R@10 |
| **Teacher Model (ViT-B/16)** | 17.80% | 34.10% | 42.44% | 14.71% | 29.87% | 38.26% |
| **Student Models (RN34)** | | | | | | |
| CL Only (Oord et al., 2018) | 4.66% | 14.10% | 21.28% | 3.78% | 11.95% | 18.40% |
| CL + MSE (Yang et al., 2024) | 4.55% | 14.27% | 21.36% | 3.79% | 11.99% | 18.44% |
| CL + KL (Li et al., 2024b) | 4.70% | 14.46% | 22.21% | 4.58% | 14.15% | 21.32% |
| CL + ICL (Yang et al., 2024) | 5.52% | 16.25% | 23.92% | 4.55% | 13.86% | 20.79% |
| CL - TE1 | 7.24% | 19.88% | 28.55% | 5.68% | 16.22% | 23.71% |
| CL - TE2 | 7.02% | 20.26% | 29.46% | 5.83% | 16.54% | 24.27% |
| CL - TE1 - TE2 | 7.44% | 20.24% | 29.01% | 5.78% | 16.35% | 23.90% |
| CL + KL + MSE - TE1 | 7.74% | 21.05% | 30.14% | 5.86% | 16.85% | 24.64% |
| CL + KL + ICL - TE1 | 7.64% | 21.07% | 30.16% | **6.20%** | **17.64%** | **25.66%** |
| CL + KL + MSE + ICL - TE1 | 7.90% | 21.49% | 30.42% | 6.15% | 17.48% | 25.35% |
| ALL Loss Function | **8.48%** | **22.02%** | **30.99%** | 6.00% | 17.01% | 24.80% |

from ViT-B/16 to RN34. In contrast, varying the MI weight $\eta$ in isolation ($\gamma = 0$) produces moderate gains over the baseline but does not surpass the best TE-regularized setting for T2I, and combining MI, MSE-$\Delta$ ($\lambda > 0$), and TE does not yield additive improvements in this sweep. Taken together, these results indicate that (i) modest TE regularization is most helpful for T2I, while (ii) stronger feature distillation (larger $\beta$) primarily benefits I2T, and careful balancing is required to avoid over-regularization when multiple auxiliary terms are enabled.

Table 4: Comparison of zero-shot retrieval performance (Recall@k) of RN34-based VLM student with teacher CLIP ViT-B/16 on MSCOCO.

| $\alpha$ | $\beta$ | $\delta$ | $\eta$ | $\lambda$ | $\gamma$ | I2T R@1 | I2T R@5 | I2T R@10 | T2I R@1 | T2I R@5 | T2I R@10 |
|---|---|---|---|---|---|---|---|---|---|---|---|
| 1 | 50 | 1 | 0 | 0 | 0 | 5.81% | 17.30% | 25.89% | 5.60% | 16.64% | 24.46% |
| 1 | 50 | 1 | 0 | 0 | 1 | 7.32% | 20.41% | 29.58% | **6.75%** | **18.74%** | **26.91%** |
| 1 | 50 | 1 | 0 | 0 | 2.5 | 7.68% | 21.16% | 30.20% | 6.70% | **18.74%** | 26.78% |
| 1 | 50 | 1 | 0 | 0 | 5 | 7.87% | 21.47% | 30.74% | 5.98% | 17.21% | 24.96% |
| 1 | 100 | 1 | 0 | 0 | 5 | **8.48%** | **22.02%** | **30.99%** | 6.00% | 17.01% | 24.80% |
| 1 | 50 | 1 | 0 | 0 | 7.5 | 7.90% | 21.27% | 30.22% | 5.92% | 16.69% | 24.42% |
| 1 | 50 | 1 | 0 | 0 | 10 | 7.49% | 20.56% | 29.51% | 5.47% | 16.01% | 23.38% |
| 1 | 50 | 1 | 1 | 0 | 0 | 7.08% | 19.72% | 28.44% | 6.40% | 17.94% | 25.86% |
| 1 | 50 | 1 | 2.5 | 0 | 0 | 7.09% | 20.00% | 29.04% | 6.31% | 17.97% | 25.96% |
| 1 | 50 | 1 | 5 | 0 | 0 | 7.28% | 20.05% | 29.24% | 6.08% | 17.37% | 25.30% |
| 1 | 100 | 1 | 5 | 0 | 0 | 7.04% | 20.17% | 29.08% | 6.21% | 17.43% | 25.37% |
| 1 | 50 | 1 | 7.5 | 0 | 0 | 6.69% | 19.53% | 28.25% | 6.01% | 16.96% | 24.70% |
| 1 | 50 | 1 | 2.5 | 0 | 5 | 7.29% | 20.45% | 29.37% | 5.68% | 16.56% | 24.13% |
| 1 | 50 | 1 | 2.5 | 50 | 2.5 | 7.33% | 20.31% | 29.40% | 6.00% | 17.12% | 24.91% |
| 1 | 50 | 1 | 1 | 0 | 7.5 | 7.43% | 20.43% | 29.56% | 5.75% | 16.48% | 24.00% |
| 1 | 50 | 1 | 1 | 50 | 7.5 | 7.42% | 20.51% | 29.25% | 5.65% | 16.33% | 24.02% |

Performance continues to improve as $\gamma$ increases up to 5, with the best I2T results observed at $\gamma = 7.5$ (7.90% Recall@1, 30.22% Recall@10). However, the T2I results peak earlier, with $\gamma = 1$ providing the strongest Recall@1 and Recall@5 values, while larger $\gamma$ values cause a mild decline. This indicates that while TE

is generally beneficial, excessively weighting it can distort the loss balance and harm retrieval performance on certain tasks. Overall, these results demonstrate two key points: (i) TE is a crucial component of the loss, consistently lifting performance above the no-TE baseline, and (ii) the optimal $\gamma$ value is task-dependent, suggesting that moderate TE weighting is sufficient to maximize the gains from information-theoretic regularization.

Table 5: Zero-shot retrieval performance (Recall@k) on Flickr8k. The RN34-based VLM student is distilled from the teacher model (CLIP ViT-B/16). All Loss Function: CL + MSE + KL + ICL - TE1 - TE2. The weighting factors for loss functions: $\alpha = 1.0$, $\beta = 100$, $\delta = 1.0$, $\gamma = 5.0$.

| Model and Loss Function | I2T Retrieval (R) | | | T2I Retrieval (R) | | |
|---|---|---|---|---|---|---|
| | R@1 | R@5 | R@10 | R@1 | R@5 | R@10 |
| **Teacher Model (ViT-B/16)** | 57.41% | 82.70% | 90.61% | 55.02% | 81.63% | 87.64% |
| **Student Models (RN34)** | | | | | | |
| CL (Oord et al., 2018) | 21.09% | 46.95% | 59.47% | 17.53% | 42.59% | 55.26% |
| CL + MSE (Yang et al., 2024) | 21.17% | 46.46% | 58.98% | 16.26% | 42.83% | 55.37% |
| CL + KL (Li et al., 2024b) | 24.38% | 51.32% | 63.84% | 19.59% | 46.97% | 60.03% |
| CL + ICL (Yang et al., 2024) | 26.44% | 52.14% | 65.32% | 20.44% | 47.69% | 61.61% |
| CL - TE1 | 28.91% | 57.17% | 69.19% | 22.98% | 50.69% | 63.79% |
| CL - TE2 | 30.07% | 57.41% | 68.45% | 23.67% | 52.04% | 65.44% |
| CL - TE1 - TE2 | 28.42% | 58.90% | 70.02% | 22.59% | 51.10% | 64.79% |
| CL + KL + MSE - TE1 | 29.90% | 59.97% | 71.83% | 22.34% | 51.73% | 65.44% |
| CL + KL + ICL - TE1 | 28.42% | 56.84% | 71.17% | 23.05% | 51.53% | 65.09% |
| CL + KL + MSE + ICL - TE1 | 29.82% | 60.71% | 71.75% | 23.53% | 51.47% | 64.99% |
| All Loss Function | **33.28%** | **64.33%** | **73.97%** | **26.36%** | **56.18%** | **69.64%** |

Flickr8k has a small number of training/evaluation samples which can amplify variance and make performance more sensitive to hyperparameter choices. Motivated by this, we additionally evaluate on Flickr30k (Young et al., 2014), a larger and more diverse captioned dataset, to provide a more robust assessment of our distillation objective and to verify that the observed benefits of TE-based regularization persist under a less data-limited setting. Table 6 summarizes the zero-shot retrieval results on Flickr30k when distilling an RN34 student from a ViT-B/16 teacher under different loss weightings. Several consistent trends emerge. First, introducing the TE reward ($\gamma > 0$) generally improves retrieval over the baseline without TE, particularly for moderate TE weights. Increasing $\gamma$ from 0 to 2.5–5 yields clear gains in both I2T and T2I Recall@K, with I2T R@1 improving from 34.71% to over 40%, and T2I R@1 rising from 29.35% to above 31%. However, very large TE weights (e.g., $\gamma = 7.5$) begin to slightly degrade performance, indicating that excessive regularization may oversuppress the contrastive signal. Second, adjusting the KL weight shows that a stronger feature-matching term ($\alpha = 5$) in combination with moderate TE ($\gamma = 5$) leads to the strongest overall I2T performance (R@1 = 41.32%, R@10 = 80.47%), suggesting complementary effects between KL alignment and TE-based guidance. In contrast, increasing $\delta$ or $\lambda$ yields smaller and less consistent gains, whereas introducing MI ($\eta > 0$) can modestly improve T2I performance. Overall, these results reinforce the pattern observed on Flickr8k: moderate TE regularization enhances the semantic structure preserved in the distilled model, while careful balancing with KL produces the most robust improvements.

## 5.3 Computational Overhead and Robustness of the Empirical Evaluation

Appendix E explains why computing exact TE is intractable, but it does not quantify the additional cost of our TE surrogates (TE1 and TE2) relative to a standard distillation run. To make this overhead explicit, we measured wall-clock time and FLOPS for two settings on Google Colab A100 GPUs: training with the baseline loss (CL + MSE + KL + ICL) and training with the objective including TE (CL + MSE + KL + ICL - TE). We report the total time for 10 training epochs plus evaluation, as well as the total FLOPS and the portion attributable to the TE terms, in Table 7.

Table 6: Comparison of zero-shot retrieval performance (Recall@k) of RN34-based VLM student with teacher CLIP ViT-B/16 on **Flickr30k**.

| $\alpha$ | $\beta$ | $\delta$ | $\eta$ | $\lambda$ | $\gamma$ | I2T R@1 | I2T R@5 | I2T R@10 | T2I R@1 | T2I R@5 | T2I R@10 |
|---|---|---|---|---|---|---|---|---|---|---|---|
| 1 | 50 | 1 | 0 | 0 | 0 | 34.71% | 62.43% | 74.85% | 29.35% | 57.91% | 70.02% |
| 1 | 50 | 1 | 0 | 0 | 1 | 38.36% | 66.67% | 76.43% | 31.40% | 60.95% | 71.93% |
| 1 | 50 | 1 | 0 | 0 | 2.5 | 40.34% | 67.16% | 77.51% | 31.79% | 60.89% | 72.01% |
| 1 | 50 | 1 | 0 | 0 | 5 | 40.24% | 68.54% | 78.80% | 31.01% | 60.37% | 72.19% |
| 1 | 50 | 1 | 0 | 0 | 7.5 | 37.28% | 66.86% | 78.11% | 30.02% | 58.93% | 71.28% |
| 1 | 50 | 1 | 0 | 0 | 10 | 39.35% | 66.67% | 76.73% | 30.20% | 58.64% | 70.43% |
| 5 | 50 | 1 | 0 | 0 | 5 | **42.31%** | **69.92%** | **80.37%** | 31.24% | 61.08% | 72.19% |
| 1 | 100 | 1 | 0 | 0 | 5 | 37.87% | 67.36% | 78.90% | 31.12% | 61.24% | 72.31% |
| 1 | 50 | 5 | 0 | 0 | 5 | 37.38% | 67.26% | 77.61% | 31.36% | 60.37% | 71.05% |
| 1 | 50 | 1 | 0 | 50 | 0 | 36.49% | 64.20% | 75.54% | 31.10% | 58.78% | 70.26% |
| 1 | 50 | 1 | 0 | 100 | 0 | 36.59% | 64.99% | 76.63% | 30.08% | 59.90% | 71.76% |
| 1 | 50 | 1 | 1 | 1 | 0 | 36.59% | 66.77% | 78.30% | 32.29% | **61.40%** | **73.06%** |
| 1 | 50 | 1 | 2.5 | 0 | 0 | 37.77% | 66.47% | 77.22% | **32.96%** | 60.79% | 71.64% |
| 1 | 50 | 1 | 5 | 0 | 0 | 39.25% | 65.78% | 76.73% | 31.52% | 60.02% | 71.91% |
| 1 | 50 | 1 | 7.5 | 0 | 0 | 39.55% | 68.05% | 78.40% | 31.95% | 60.00% | 71.48% |
| 5 | 50 | 1 | 7.5 | 0 | 0 | 37.77% | 67.36% | 77.91% | 30.14% | 58.21% | 69.92% |
| 1 | 50 | 1 | 2.5 | 50 | 2.5 | 40.43% | 67.06% | 78.11% | 32.64% | 60.65% | 72.21% |
| 1 | 50 | 1 | 2.5 | 0 | 5 | 38.66% | 66.77% | 78.80% | 30.51% | 59.47% | 71.95% |
| 1 | 50 | 1 | 2.5 | 50 | 5 | 39.94% | 68.84% | 78.40% | 31.32% | 59.68% | 71.34% |
| 1 | 50 | 1 | 1 | 0 | 7.5 | 36.39% | 67.65% | 77.91% | 30.73% | 59.39% | 71.48% |
| 1 | 50 | 1 | 1 | 50 | 7.5 | 38.17% | 65.78% | 77.02% | 30.08% | 58.09% | 70.91% |
| 1 | 50 | 1 | 5 | 50 | 7.5 | 36.79% | 65.48% | 76.82% | 30.41% | 58.86% | 70.49% |
| **Teacher (ViT-B/16)** | | | | | | 81.76% | 97.14% | 98.92% | 62.66% | 86.23% | 91.36% |

Table 7: Computation and time cost comparison for training (10 epochs) and evaluation on MSCOCO.

| Teacher | Student | CL+KL+MSE+ICL–TE | CL+KL+MSE+ICL | FLOPS | FLOPS from TE |
|---|---|---|---|---|---|
| RN50 | RN34 | 6h 19m | 6h 18m | $3.44 \times 10^{16}$ | $1.79 \times 10^{10}$ |
| ViT-B/16 | RN34 | 5h 33m | 5h 32m | $2.60 \times 10^{16}$ | $1.49 \times 10^{10}$ |

The reported FLOPS correspond to the entire training and evaluation run with the loss CL+KL+MSE+ICL-TE. The FLOPS contributed by the TE surrogates are six orders of magnitude smaller than the total, and the wall-clock times with and without TE only have 1 minute difference, indicating that the additional computational cost of TE1/TE2 is negligible in practice. The experiment with teacher ViT-B/16 is faster because its image encoder has hidden dimension 768, whereas RN50 uses hidden dimension 1024.

To assess the robustness of our empirical findings, we run each configuration with five independent random seeds and report mean ± standard deviation for all retrieval metrics in Table 8. On MSCOCO, the standard deviations are very small (typically on the order of 0.1–0.3), indicating that both I2T and T2I performance are highly stable under different initializations and data shuffling. Flickr30k exhibits slightly larger but still modest variability, suggesting mild sensitivity while remaining well-behaved. In contrast, Flickr8k shows noticeably larger standard deviations (around 1.0), which is expected given its much smaller size and higher sampling noise. Overall, these trends confirm that our conclusions are robust on larger benchmarks, with variability naturally increasing on smaller datasets such as Flickr8k.

Beyond Recall@K, we also evaluate Mean Reciprocal Rank (MRR) Schütze et al. (2008) for both I2T and T2I retrieval in Table 9. For I2T, each image is associated with multiple ground-truth captions: we compute similarities to all caption embeddings, determine the 1-indexed rank of each correct caption, and use the best

Table 8: Zero-shot retrieval performance (mean ± std, %) of RN34-based VLM students distilled from different CLIP teachers on multiple datasets ($\delta = 1, \lambda = 0$).

| Teacher | Dataset | $\alpha$ | $\beta$ | $\eta$ | $\gamma$ | I2T Recall@K | | | T2I Recall@K |
|---|---|---|---|---|---|---|---|---|---|
| | | | | | | R@1 | R@5 | R@10 | R@1 / R@5 / R@10 |
| RN50 | MSCOCO | 1 | 50 | 0 | 7.5 | 10.08±0.17 | 26.10±0.19 | 36.22±0.21 | 7.12±0.19 / 19.35±0.27 / 27.85±0.23 |
| RN50 | MSCOCO | 1 | 50 | 5 | 0 | 8.35±0.22 | 23.34±0.16 | 33.25±0.24 | 7.35±0.28 / 20.57±0.18 / 28.95±0.26 |
| RN50 | Flickr8k | 1 | 50 | 0 | 5 | 35.81±1.12 | 65.67±1.13 | 77.47±1.19 | 26.74±1.11 / 56.02±1.03 / 69.22±1.02 |
| RN50 | Flickr8k | 1 | 50 | 5 | 0 | 33.22±1.15 | 64.18±1.09 | 76.90±1.08 | 26.32±1.21 / 57.12±1.12 / 69.31±1.06 |
| ViT-B/16 | MSCOCO | 1 | 100 | 0 | 5 | 8.01±0.21 | 21.71±0.18 | 30.75±0.16 | 6.03±0.23 / 17.27±0.15 / 25.03±0.18 |
| ViT-B/16 | MSCOCO | 1 | 50 | 5 | 0 | 7.07±0.16 | 19.98±0.22 | 28.97±0.17 | 6.13±0.26 / 17.46±0.16 / 25.30±0.21 |
| ViT-B/16 | Flickr30k | 5 | 50 | 0 | 5 | 42.24±0.79 | 69.42±0.84 | 79.84±0.89 | 31.46±0.82 / 60.56±0.69 / 72.11±0.62 |
| ViT-B/16 | Flickr30k | 1 | 50 | 5 | 0 | 38.75±0.81 | 65.92±0.78 | 76.23±0.83 | 31.21±0.72 / 60.14±0.64 / 71.70±0.68 |

(minimum) rank to accumulate the reciprocal rank 1/rank. For T2I, each caption has a single correct image; we rank all images by similarity to the caption and again use 1/rank to compute MRR. All metrics are computed over the full validation split so that MRR follows the same protocol as Recall@K while providing a more fine-grained assessment of ranking quality across the entire list.

Table 9: Comparison of zero-shot MRR (mean ± std) with different settings ($\delta = 1$, $\lambda = 0$).

| Teacher | Student | Dataset | $\alpha$ | $\beta$ | $\eta$ | $\gamma$ | I2T MRR (%) | T2I MRR (%) |
|---|---|---|---|---|---|---|---|---|
| RN50 | RN34 | MSCOCO | 1 | 50 | 0 | 7.5 | 18.32±0.12 | 13.93±0.19 |
| RN50 | RN34 | MSCOCO | 1 | 50 | 5 | 0 | 16.46±0.14 | 14.34±0.23 |
| RN50 | RN34 | Flickr8k | 1 | 50 | 0 | 5 | 49.03±1.02 | 39.09±1.15 |
| RN50 | RN34 | Flickr8k | 1 | 50 | 5 | 0 | 44.62±1.17 | 38.47±1.12 |
| ViT-B/16 | RN34 | MSCOCO | 1 | 100 | 0 | 5 | 15.52±0.15 | 12.46±0.11 |
| ViT-B/16 | RN34 | MSCOCO | 1 | 50 | 5 | 0 | 14.23±0.22 | 12.63±0.09 |
| ViT-B/16 | RN34 | Flickr30k | 5 | 50 | 0 | 5 | 54.79±0.74 | 45.38±0.65 |
| ViT-B/16 | RN34 | Flickr30k | 1 | 50 | 5 | 0 | 52.21±0.62 | 45.02±0.71 |

Table 8 and Table 9 also compare the TE-based distillation term (settings with $\eta = 0$, $\gamma > 0$) against the MI-based baseline ($\eta > 0$, $\gamma = 0$). Across all four teacher–student–dataset combinations, TE consistently achieves higher I2T performance than MI, both in Recall@K and in MRR. For T2I, TE is competitive with or better than MI on most metrics, with MI only occasionally matching or slightly exceeding TE on a single Recall@K value. On the larger MSCOCO and Flickr30k benchmarks, the gains from TE are especially clear, indicating that the TE surrogate provides a stronger and more reliable signal for shaping the multimodal embedding space. On the smaller Flickr8k dataset, TE still yields higher I2T MRR and Recall@K than MI despite the higher variance, suggesting that geometric alignment of teacher–student updates is more effective than MI-based matching under capacity mismatch. Overall, TE outperforms MI in our settings, improving the ranking quality of the student model while maintaining stable training dynamics.

## 5.4 More Competitive Teacher Model

Table 10 reports zero-shot retrieval results when distilling a student RN34 from a substantially more competitive teacher, RN50×16, on MSCOCO. Across the explored settings, the student achieves consistent improvements over the RN50/RN34 and ViT-B/16/RN34 pairings, suggesting that a stronger teacher provides richer supervisory signals even under the same distillation objective. We observe that moderate TE regularization remains beneficial: with $\gamma \in \{2.5, 5\}$ the student attains the best overall trade-offs, achieving the strongest T2I performance at $\gamma = 2.5$ (R@1/5/10 of 8.40/22.45/31.50) and the strongest I2T performance when increasing the feature distillation weight to $\beta = 100$ with $\gamma = 5$ (R@1/5/10 of 10.75/27.20/37.41). Larger TE weights (e.g., $\gamma = 7.5$) slightly reduce both I2T and T2I, again indicating diminishing returns and potential over-regularization when the reward term is too strong. Introducing the MI term alone ($\eta \in \{1, 2.5, 5\}$ with $\gamma = 0$) yields competitive but slightly lower results than the best TE-regularized configurations, and combining additional auxiliary terms (MI and MSE-$\Delta$) does not produce clear additive gains within this

sweep. Finally, despite the strong teacher performance (I2T R@1 of 32.14%; T2I R@1 of 17.08%), a sizable teacher-student gap remains, highlighting the difficulty of compressing RN50×16 into RN34 while still demonstrating that carefully tuned TE and feature distillation substantially improve the student's retrieval accuracy.

Table 10: Comparison of zero-shot retrieval performance (Recall@k) in percentage of RN34-based VLM student with teacher CLIP RN50×16 on MSCOCO.

| $\alpha$ | $\beta$ | $\delta$ | $\eta$ | $\lambda$ | $\gamma$ | I2T R@1 | I2T R@5 | I2T R@10 | T2I R@1 | T2I R@5 | T2I R@10 |
|---|---|---|---|---|---|---|---|---|---|---|---|
| 1 | 50 | 1 | 0 | 0 | 2.5 | 10.29% | 26.18% | 36.56% | **8.40%** | **22.45%** | **31.50%** |
| 1 | 50 | 1 | 0 | 0 | 5 | 10.52% | 26.88% | 37.19% | 7.85% | 21.11% | 29.80% |
| 1 | 100 | 1 | 0 | 0 | 5 | **10.75%** | **27.20%** | **37.41%** | 8.01% | 21.32% | 29.86% |
| 1 | 50 | 1 | 0 | 0 | 7.5 | 9.99% | 26.47% | 36.48% | 7.66% | 20.51% | 29.06% |
| 1 | 50 | 1 | 1 | 0 | 0 | 9.64% | 25.14% | 35.55% | 7.64% | 20.97% | 29.79% |
| 1 | 50 | 1 | 2.5 | 0 | 0 | 9.93% | 25.94% | 35.97% | 7.84% | 21.26% | 30.22% |
| 1 | 50 | 1 | 5 | 0 | 0 | 9.81% | 25.57% | 35.87% | 7.79% | 20.99% | 29.80% |
| 1 | 50 | 1 | 2.5 | 50 | 5 | 10.01% | 25.97% | 36.29% | 7.77% | 20.88% | 29.58% |
| 1 | 50 | 1 | 1 | 50 | 7.5 | 10.08% | 26.00% | 36.21% | 7.49% | 20.21% | 28.73% |
| **Teacher (RN50×16)** | | | | | | 32.14% | 53.62% | 62.67% | 17.08% | 33.54% | 42.00% |

## 5.5 Additional Experiments

In Appendix H, we present the experimental results for five more experiments: 1) teacher: RN50, student: RN34, dataset: Flick8k; 2) teacher: RN50, student: RN18, dataset: MSCOCO; 3) teacher: RN50, student: RN34, dataset: Food-101, for classification; 4) teacher: RN50, student: RN34, Datasets: Food-101 & MSCOCO for distillation and ImageNet-1k validation dataset for zero-shot classification; 5) qualitative embedding-space visualization.

They provide additional evidence of the generality of our TE-guided distillation across student capacities and downstream tasks. Finally, to assess out-of-domain semantic retention, we report zero-shot ImageNet-1k (50K val images) results for the RN34 distilled on MSCOCO+Food-101; moderate TE weighting ($\gamma \in \{5, 7.5\}$) yields the strongest ImageNet accuracies and outperforms MI-only variants, although absolute performance remains substantially below the teacher due to the student's smaller capacity and the substantial distribution shift. It also provides a qualitative view of how the student and teacher organize the CLIP embedding space after distillation.

## 6 Conclusions and Future Work

In this work, we introduced TE as a regularization technique for VLM distillation, aiming to enhance knowledge transfer from a teacher model to a student model. Direct computation of TE is intractable due to the high dimensionality of image and text representations. To address this, we demonstrated that a first-order (linear) expansion of TE yields a practical surrogate based on the cosine similarity between the Jacobians of the teacher and student processes. Building on this insight, we proposed two TE approximation strategies that leverage cosine similarity to quantify and enforce directional information flow between teacher and student embeddings across both image and text modalities. By integrating TE-based regularization into the distillation loss, we showed that the student model more effectively captures structured multimodal information, resulting in improved retrieval performance.

Our experiments were conducted using CLIP RN50, ViT-B/16, RN50×16 as teacher models, and RN34 and RN18 as student models, evaluated on the MSCOCO 2014, Flickr8k, Flickr30k, Food-101, ImageNet-1k datasets. The experimental results underscore the importance of TE-based regularization for achieving improved feature alignment. Student models trained with TE consistently outperform those trained without TE, exhibiting notable gains in Recall@k for both image-to-text and text-to-image retrieval tasks.

This work primarily focuses on static teacher-student distillation, where the teacher model remains fixed during training. Future directions include extending our approach to co-distillation scenarios, wherein both teacher and student are jointly optimized to mutually enhance knowledge transfer. Additionally, exploring TE-based reinforcement learning strategies may provide an alternative optimization framework, enabling the student model to actively maximize meaningful information flow throughout training.

## 7    Limitations

Despite the demonstrated effectiveness of TE-based regularization in VLM distillation, our approach has several limitations. First, the approximation of transfer entropy in high-dimensional embedding spaces via cosine similarity, while computationally efficient, may not fully capture the complex nonlinear dependencies present in deep representations. Second, our surrogate TE formulations focus on global embedding statistics and directional changes, which might overlook fine-grained or instance-specific information transfer between teacher and student. Third, the current framework assumes availability of paired teacher-student representations for each sample, which may not generalize to settings with partial supervision or noisy teacher signals.

While TE1/TE2 consistently improve retrieval, the student can still remain far from the teacher because our surrogates capture only a *restricted* geometric signal: alignment of *local* representation variations (finite-difference directions) between teacher and student on a batch. Several important aspects of the teacher geometry are not directly constrained. First, TE1/TE2 do not enforce *global* isometry of the embedding space (e.g., preservation of long-range pairwise distances, neighborhood structure across the full data distribution, or higher-order manifold curvature), so a student may match local directional trends while still being globally mis-scaled or warped. Second, the surrogates use *directional* alignment but do not control the *magnitude* or anisotropy spectrum of variation (e.g., singular-value profile / principal directions), which can affect retrieval even when cosine alignment is high. Third, TE1/TE2 operate at the level of final embeddings and thus do not capture mismatches in *intermediate* representations, cross-modal interaction mechanisms, or capacity-dependent inductive biases that make the teacher's embedding space more semantically separable. Finally, because the differences are formed from within-batch pairings, the surrogate only samples a small subset of directions in input space; directions that are important for hard negatives, rare concepts, or fine-grained attributes may be underrepresented, leaving residual error. These limitations suggest that TE1/TE2 are best viewed as efficient geometric regularizers that complement (rather than replace) distributional matching losses (e.g., KL/contrastive/ICL), and that closing the remaining gap likely requires stronger capacity, richer direction sampling, or additional global-structure constraints.

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

## A    Relations between Transfer Entropy, Entropy, and Mutual Information

For two discrete-time stochastic processes $X(t)$ and $Y(t)$, the transfer entropy from $X$ to $Y$ is formally defined as (Schreiber, 2000):

$$T_{X \to Y} = \sum_t p(y_{t+1}, y_t, x_t) \log \frac{p(y_{t+1} \mid y_t, x_t)}{p(y_{t+1} \mid y_t)}, \tag{21}$$

where $p(\cdot)$ represents probability distributions of the respective random variables.

Transfer entropy can also be expressed in terms of conditional entropy and mutual information (Shahsavari Baboukani et al., 2020). Specifically, the transfer entropy from $X$ to $Y$, denoted $T_{X \to Y}$, measures the reduction in uncertainty about the future state $Y_{t+1}$ given the joint past of $X$ and $Y$, compared to the uncertainty given the past of $Y$ alone. Based on (21), this difference can be expressed as:

$$
\begin{aligned}
T_{X \to Y} &= \sum_t p(y_{t+1}, y_t, x_t) \log \frac{1}{p(y_{t+1} \mid y_t)} + \\
&\quad \sum_t p(y_{t+1}, y_t, x_t) \log p(y_{t+1} \mid y_t, x_t) \\
&= \sum_t p(y_{t+1}, y_t) \log \frac{1}{p(y_{t+1} \mid y_t)} + \\
&\quad \sum_t p(y_{t+1}, y_t, x_t) \log p(y_{t+1} \mid y_t, x_t) \\
&= H(Y_{t+1} \mid Y_t) - H(Y_{t+1} \mid Y_t, X_t) \\
&= I(Y_{t+1}; X_t \mid Y_t)
\end{aligned}
$$

$$\text{(22)}$$
$$\text{(23)}$$

where $H(Y_{t+1} \mid Y_t)$ is the conditional entropy of $Y_{t+1}$ given its own history $Y_t$, $H(Y_{t+1} \mid Y_t, X_t)$ is the conditional entropy of $Y_{t+1}$ given both the history of $Y$ and the history of $X$, and $I(Y_{t+1}; X_t \mid Y_t)$ is the mutual information between $Y_{t+1}$ and the history of $X$, conditioned on the history of $Y$. In this formulation, the transfer entropy quantifies the amount by which the uncertainty about the future of $Y$ is reduced by incorporating information from $X$.

## B    Prior Work on Mutual Information and Transfer Entropy Estimation

Mutual Information (MI) techniques have been employed to capture shared information between variables (Hjelm et al., 2018)(Oord et al., 2018)(Tian et al., 2020). MINE (Belghazi et al., 2018) offers a differentiable estimator for mutual information, and information-theoretic regularization has been applied in generative models for disentanglement and improved control (Chen et al., 2016). In (Gao et al., 2015), a mutual information estimator was proposed based on modified k-nearest neighbor (KNN) that is robust to local non-uniformity with limited data. A diverse set of distributions with known MI values were introduced to evaluate the performance of different MI estimators beyond traditional normal distributions (Czyż et al., 2023). McAllester and Stratos (McAllester & Stratos, 2020) highlighted the inherent difficulties in estimating mutual information from finite data, demonstrating that any distribution-free high-confidence lower bound on MI cannot exceed $O(\ln N)$, thereby underscoring the fundamental challenges in accurate mutual information estimation without strong assumptions about the data distribution. Goldfeld and Greenewald (Goldfeld & Greenewald, 2021) introduced Sliced Mutual Information, a scalable measure that projects high-dimensional distributions onto one-dimensional subspaces, effectively capturing complex dependencies while reducing computational complexity. Approximating mutual information of high-dimensional variables using learned representations was studied in (Gowri et al., 2025).

Transfer entropy is a conditional mutual information from two stochastic processes, so it's more challenging in TE estimation. In (Zhang, 2018), Low-dimensional approximation in the searching procedure was applied to transfer entropy from non-uniform embedding. In (Zhu et al., 2015), KNN was used for TE estimation. However, KNN-based approach doesn't work well if the data are noisy and long ranged. To

overcome this weakness, a perturbation model based on locality sensitive hash function was proposed for TE estimation (Garg et al., 2022). Three estimators were used for TE estimation (Lee et al., 2012), namely fixed-binning with ranking, kernel density estimation, and the Darbellay-Vajda (D-V) adaptive partitioning algorithm extended to three dimensions. In (Ma, 2019), copula entropy was applied to TE estimation. To overcome the curse of dimensionality in TE estimation, TE was decomposed into a sum of finite-dimensional contributions in (Runge et al., 2012). Recently, transformer was used for TE estimation (Luxembourg et al., 2024). In this paper, we propose TE approximation approaches which can tremendously reduce the computation cost and overcome the curse of dimensionality.

## C  Proof of Theorem 1

This section shows how a first-order (linear) expansion of TE leads to a computable surrogate based on a cosine similarity between the teacher– and student-process Jacobians. Our derivation follows the *linear–Gaussian surrogate* technique proposed in (Goldfeld et al., 2019).

*Proof.* Let $x \in \mathbb{R}^d$ be an input image–caption pair, and $f_T(x)$, $f_S(x) \in \mathbb{R}^D$ denote the teacher and student embeddings, respectively. Denote their Jacobians as $J_T(x) = \nabla_x f_T(x)$ and $J_S(x) = \nabla_x f_S(x)$, both in $\mathbb{R}^{D \times d}$.

To study the local behavior around $x$, consider a small perturbation $\delta x \sim \mathcal{N}(0, \sigma^2 I_d)$, and define

$$u := f_T(x + \delta x), \qquad v_t := f_S(x), \qquad v_{t+1} := f_S(x + \delta x).$$

The one-step transfer entropy from teacher to student becomes:

$$T_T^S(x) = I\big(v_{t+1};\, u \,|\, v_t\big). \tag{24}$$

Using a first-order Taylor expansion around $x$:

$$u \approx u_0 + J_T \delta x, \quad v_{t+1} \approx v_0 + J_S \delta x, \quad v_t = v_0 = f_S(x), \tag{25}$$

where $u_0 = f_T(x)$. Since $v_0$ is a constant shift, subtracting it from both sides does not change the conditional mutual information. Therefore:

$$T_T^S(x) \approx I\big(J_S \delta x;\, J_T \delta x\big). \tag{26}$$

Because $\delta x \sim \mathcal{N}(0, \sigma^2 I_d)$ and both $J_S$ and $J_T$ are linear maps, the pair $(J_S \delta x, J_T \delta x)$ is jointly Gaussian. Define the covariances:

$$\Sigma_S = \sigma^2 J_S J_S^\top, \quad \Sigma_T = \sigma^2 J_T J_T^\top, \quad \Sigma_{ST} = \sigma^2 J_S J_T^\top.$$

The mutual information between jointly Gaussian vectors is (Cover, 1999):

$$
\begin{aligned}
I\big(J_S \delta x;\, J_T \delta x\big) &= h\big(J_S \delta x\big) + h\big(J_T \delta x\big) - h\big(J_S \delta x,\, J_T \delta x\big) \tag{27}\\
&= \frac{1}{2} \log \frac{\det \Sigma_S \, \det \Sigma_T}{\det \begin{pmatrix} \Sigma_S & \Sigma_{ST} \\ \Sigma_{TS} & \Sigma_T \end{pmatrix}} \tag{28}\\
&= -\frac{1}{2} \log \det \Big(I - \Sigma_S^{-1/2} \Sigma_{ST} \Sigma_T^{-1/2}\Big). \tag{29}
\end{aligned}
$$

If $\Sigma_{ST}$ is small compared to the product $\Sigma_S^{1/2} \Sigma_T^{1/2}$ (which is often true in early training), we can use the approximation $\log \det(I - A) \approx -\operatorname{tr}(A)$ (Magnus & Neudecker, 1999). This gives (Goldfeld et al., 2019):

$$T_T^S(x) \approx \frac{\sigma^2}{2} \operatorname{tr}\left((J_S J_S^\top)^{-1/2} J_S J_T^\top (J_T J_T^\top)^{-1/2}\right). \tag{30}$$

We can normalize both Jacobians by their Frobenius norms:

$$\widetilde{J}_S = \frac{J_S}{\|J_S\|_F}, \qquad \widetilde{J}_T = \frac{J_T}{\|J_T\|_F},$$

so that equation equation 30 becomes:

$$T_T^S(x) \propto \langle \widetilde{J}_S, \ \widetilde{J}_T \rangle_F = \cos\left(\widetilde{J}_S, \ \widetilde{J}_T\right), \tag{31}$$

i.e., the Frobenius inner product (cosine similarity) of the two Jacobians.

$\square$

## D  Performance Comparison: TE Approximations versus Exact TE

We evaluated our two approximations of TE in Section 4.2 against the exact TE computed from a synthetic Gaussian channel. Specifically, teacher embeddings $\mathbf{T} \in \mathbb{R}^D$ are sampled from a standard normal distribution $\mathbf{T} \sim \mathcal{N}(0, I)$ , and student embeddings are generated as

$$\mathbf{S} = \alpha \mathbf{T} + \sqrt{1 - \alpha^2} \mathbf{N}, \tag{32}$$

where $\mathbf{N} \sim \mathcal{N}(0, I)$ and $\alpha \in [0, 0.99]$ controls the teacher–student correlation. So each corresponding pair of teacher and student components forms a jointly Gaussian random pair with Pearson correlation coefficient $\alpha$ (Lee Rodgers & Nicewander, 1988). It is a classical result in information theory that for two jointly Gaussian random variables $X$ and $Y$ with correlation $\alpha$, the mutual information is given by (Cover, 1999)

$$I(X; Y) = -\frac{1}{2} \log\left(1 - \alpha^2\right). \tag{33}$$

In our setting, the exact transfer entropy is defined as (Shahsavari Baboukani et al., 2020)

$$\text{TE}_{\text{exact}} = I(Y_{t+1}; X_t \mid Y_t), \tag{34}$$

where $Y_{t+1}$ represents the student's updated representation, $X_t$ is the teacher's representation at time $t$, and $Y_t$ is the student's current representation. Under the common assumption that these variables are jointly Gaussian and the update of $Y_{t+1}$ depends linearly on $X_t$ (after conditioning on $Y_t$), a closed-form expression for the conditional mutual information can be derived. In particular, if the effective correlation between $X_t$ and $Y_{t+1}$ (after accounting for $Y_t$) is given by $\alpha$, then the mutual information per embedding dimension becomes

$$I(Y_{t+1}; X_t \mid Y_t) = -\frac{1}{2} \log\left(1 - \alpha^2\right). \tag{35}$$

When the embeddings have $D$ independent dimensions, this yields

$$\text{TE}_{\text{exact}} = \frac{D}{2} \log\left(\frac{1}{1 - \alpha^2}\right). \tag{36}$$

For ease of comparison with our cosine similarity–based approximations, we further normalize this exact TE value via a logarithmic transformation to map it into the interval $[0, 1]$ using the following transformation (Han et al., 2012):

$$\text{TE}_{\text{norm}} = \frac{\log(1 + \text{TE}_{\text{exact}})}{\log(1 + \text{TE}_{\text{max}})}, \tag{37}$$

where $\text{TE}_{\text{max}}$ is computed using $\alpha_{\text{max}} = 0.99$ to define the upper bound for normalization.

For the two approximation methods proposed in Sections 4.2.1 (Method 1) and 4.2.2 (Method 2), we conducted experiments by varying $\alpha$ from 0 to 0.99, and computed the two approximation results and the normalized exact TE. The results are summarized in Fig. 2. The Pearson correlation between the normalized exact TE and both TE approximations was found to be 0.994, indicating a very strong linear relationship. These findings suggest that both approximation methods reliably track the exact TE, capturing the relative information flow from the teacher to the student in this synthetic setting.

We also examined the robustness of our two TE approximation methods as we varied two key factors in a synthetic teacher–student setting:

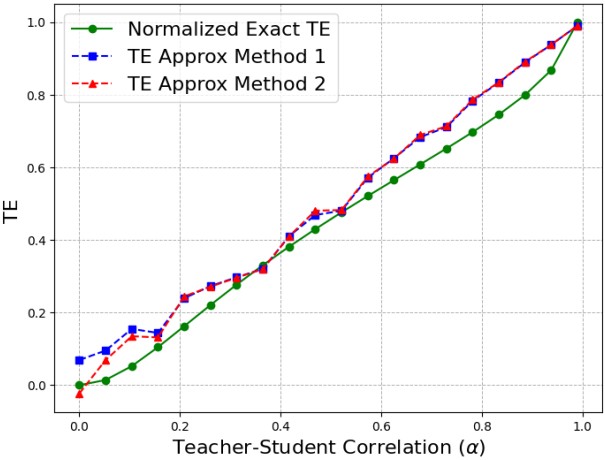

Figure 2: Comparison of TE Approximations vs. Normalized Exact TE.

- Batch size ($B$), which affects the stability of sample-based estimates.

- Embedding dimension ($D$), which influences the amount of representational capacity.

We fixed the teacher–student correlation coefficient at $\alpha = 0.8$ in (32). Two separate experiments were performed:

1. Varying batch size: We fix $D = 500$ and consider batch sizes $B \in \{10, 20, 50, 100, 200, 500, 1000\}$.

2. Varying embedding dimension: We fix $B = 500$ and let $D \in \{10, 20, 50, 100, 200, 500, 1000\}$.

In both cases, we computed the TE Approximation Method 1 and Method 2, and the normalized exact TE.

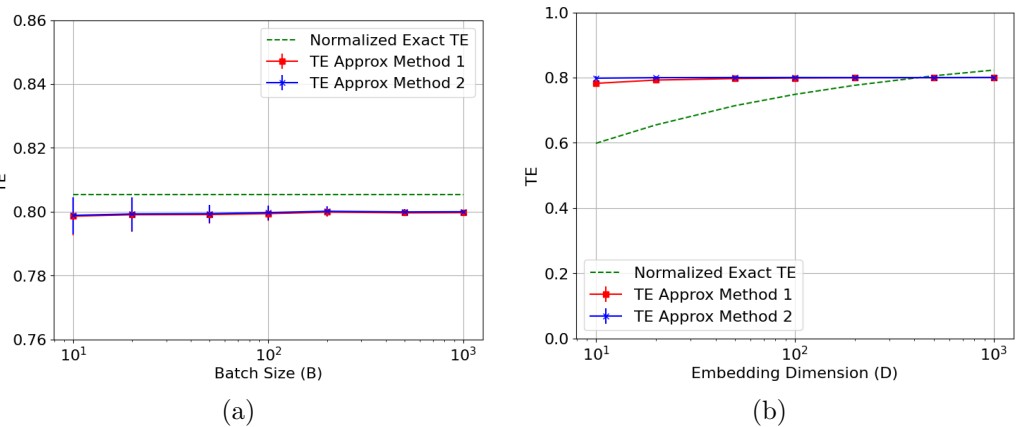

Figure 3: (a) TE approximations versus batch size ($B$) at fixed $D = 500$. (b) TE approximations versus embedding dimension ($D$) at fixed $B = 500$.

Figure 3a shows the behavior of these metrics as a function of batch size. Observe that both approximation methods rapidly converge to a stable estimate near the normalized exact TE (green dashed line). For very small $B$ (around 10–20), the sample-based cosine measures show slight deviations but still remain close to the exact TE. As $B$ grows, the variance diminishes and both approximations tightly match the theoretical reference.

Figure 3b illustrates the impact of varying embedding dimension $D$. Since the *exact* TE increases with $D$ (due to additional degrees of freedom), its normalized value (green line) also increases. By contrast, the two TE approximations remain relatively stable, hovering around 0.75–0.80 for all tested dimensions. This highlights a key property of the approximate measures: they capture the relative alignment between teacher and student (controlled by $\alpha$), but they do not grow with the embedding dimensionality as the exact mutual information does. In practice, this makes them computationally efficient and robust to high-dimensional data, though they are not designed to quantify the absolute amount of information transferred. Overall, these results confirm that both approximation methods track the ground-truth TE trend (in terms of relative comparisons), while offering a simpler and more scalable alternative to exact TE in high-dimensional settings.

The underlying intuition behind these approximation is that if the student's directional changes closely match those of the teacher, then information transfer is effectively occurring. Traditional transfer entropy measures rely on probability distributions over time, but this approach circumvents such computational overhead by leveraging geometric similarity in embedding space. By treating the batch as a sequence of evolving representations, the method estimates how well the teacher's influence propagates to the student. However, unlike traditional TE, which explicitly models information transfer through probability distributions, our approximation purely relies on directional alignment. Additionally, equal weighting of image and text modalities may not always be ideal, as one modality may contribute more to the learning process than the other.

The cosine-based TE approximations are highly effective in capturing the relative information flow in VLM distillation – they are easy to compute, robust across high dimensions, and correlate well with true information transfer. They confirm the intuition that a student embedding space matching the teacher's geometry is a good sign of successful knowledge distillation. However, these approximations do not measure exact information volume. They compress the notion of "how much knowledge" into a bounded similarity score. As a result, they are best used for comparing models or monitoring training (where the scale can be assumed fixed and only relative changes matter) rather than for absolute information quantification.

## E   Computational Cost Analysis: Exact TE versus TE Approximations

**Computational Complexity:**

Exact TE often involves $O(N^2 d)$ operations, where $N$ is the number of samples and $d$ is the feature dimension. This is due to the need for joint probability estimation over multiple variables, which scales poorly as data size and dimensionality increase. In contrast, TE Approximation Method 1 in Section 4.2.1 uses cosine similarity to estimate TE by focusing on local neighborhoods (difference of neighbors) in the embedding space. Instead of constructing a full joint probability table, for each observation one can find a set of "similar" past states (e.g. nearest neighbors in terms of cosine distance) and approximate conditional probabilities from those neighbors. The neighbor-based cosine similarity approximation reduces complexity by considering only local neighborhoods in the embedding space. By focusing on a limited number of similar past states instead of the entire dataset, this method lowers the computational cost to approximately $O(N \log N)$ with efficient neighbor searches. TE Approximation Method 2 in 4.2.2 uses cosine similarity to concatenate the high-dimensional states, thereby reducing the state space before computing TE. The concatenation-based approximation further reduces complexity by grouping similar data points into clusters and treating each cluster as a discrete state, leading to an effective time complexity of $O(Nd)$ for the clustering process and $O(N)$ for TE calculation.

**Memory Usage:** Exact TE requires storing large joint probability distributions, which grow exponentially with dimensionality. This makes exact computation infeasible for high-dimensional embeddings, as it demands large storage space for probability tables or expensive nearest-neighbor searches. The approximations mitigate this issue by avoiding explicit density estimation. The TE Approximation Method 1 only stores similarity measures and a small set of neighbors for each data point, keeping memory usage at $O(Nd)$. The TE Approximation Method 2 concatenates data into a limited number of clusters, further reducing storage requirements to $O(Cd)$, where $C$ is the number of clusters, much smaller than $N$. These approximations thus enable TE computation in large-scale deep learning applications without overwhelming memory constraints.

**Scalability in High Dimensions:** Exact TE suffers from the curse of dimensionality. As dimensionality increases, joint probability estimation becomes unreliable because high-dimensional data points become sparse, making density estimation difficult. This often results in TE values that are biased towards zero. In contrast, cosine similarity-based methods are much more scalable, as cosine similarity is well-defined even in high dimensions and can be computed efficiently. The TE Approximate method 1 relies on approximate nearest-neighbor searches, which remain feasible even as $d$ grows, while the TE Approximate method 2 concatenates high-dimensional data into a manageable number of clusters, making TE estimation practical even for very large embeddings.

In summary, using cosine similarity approximations for transfer entropy enables analysis of high-dimensional and large-scale data that would be otherwise impossible with exact methods. The approaches significantly improve computational feasibility and can even enhance statistical reliability in data-limited situations (Zhang, 2018). The cost, however, is that we must accept an approximate measure that may overlook complex nuances of the data's information dynamics. Since the priority in CLIP is to handle very rich embeddings and get a fast, actionable estimate of information flow, the TE approximation methods are invaluable.

## F  Loss Functions in VLM Distillation

### F.1  Logit Representation in VLM Distillation

In our framework, logits represent the similarity scores between image and text embeddings, which are fundamental to contrastive learning. Given a batch of image-text pairs, let $\mathbf{v}^{(S)}, \mathbf{u}^{(S)}$ denote the image and text embeddings from the student model, and $\mathbf{v}^{(T)}, \mathbf{u}^{(T)}$ denote the corresponding embeddings from the teacher model. The logit computation follows these steps.

First, we normalize the embeddings to unit norm:

$$\hat{\mathbf{v}}^{(S)} = \frac{\mathbf{v}^{(S)}}{\|\mathbf{v}^{(S)}\|_2}, \quad \hat{\mathbf{u}}^{(S)} = \frac{\mathbf{u}^{(S)}}{\|\mathbf{u}^{(S)}\|_2}, \tag{38}$$

$$\hat{\mathbf{v}}^{(T)} = \frac{\mathbf{v}^{(T)}}{\|\mathbf{v}^{(T)}\|_2}, \quad \hat{\mathbf{u}}^{(T)} = \frac{\mathbf{u}^{(T)}}{\|\mathbf{u}^{(T)}\|_2}. \tag{39}$$

The similarity logits for the student and teacher models are then computed as the dot product between the corresponding image and text embeddings, scaled by a temperature parameter $\tau$:

$$\mathbf{z}^{(S)} = \frac{\hat{\mathbf{v}}^{(S)} \cdot (\hat{\mathbf{u}}^{(S)})^\top}{\tau}, \quad \mathbf{z}^{(T)} = \frac{\hat{\mathbf{v}}^{(T)} \cdot (\hat{\mathbf{u}}^{(T)})^\top}{\tau}. \tag{40}$$

Here, $\mathbf{z}^{(S)}$ and $\mathbf{z}^{(T)}$ are $|B| \times |B|$ matrices, where each entry $z_{ij}^{(S)}$ represents the similarity between the $i$-th image embedding and the $j$-th text embedding in the batch for the student model, and similarly for the teacher model. The temperature parameter $\tau$ controls the sharpness of the similarity distribution, with lower values making the distribution more peaky.

These logits are subsequently used in the contrastive loss and KL divergence computation to align the student's feature representations with those of the teacher, ensuring effective knowledge transfer during distillation. Several studies have explored the computation and utilization of these logits in image-text contrastive frameworks (Radford et al., 2021)(Jia et al., 2021)(Yang et al., 2022)(Hasegawa et al., 2023)(Xiao et al., 2024).

### F.2  Contrastive Loss for VLM Distillation

We employ a contrastive loss based on the InfoNCE loss formulation to align the student model's image and text representations effectively. Given a batch of $|B|$ image-text pairs, we define the contrastive loss using

the computed logits. The contrastive loss for image-to-text alignment is defined as (Oord et al., 2018):

$$\mathcal{L}_{I \to T} = -\frac{1}{|B|} \sum_{k=1}^{|B|} \log \frac{\exp(z_{kk}^{(S)})}{\sum_{j=1}^{|B|} \exp(z_{kj}^{(S)})} \tag{41}$$

$z_{kj}^{(S)}$ represents the similarity between the $k$-th image embedding and the $j$-th text embedding in the batch for the student model, and $z_{kk}^{(S)}$ represents the similarity logit between the $k$-th image and its corresponding text in the batch for the student model.

Similarly, the contrastive loss for text-to-image alignment is given by:

$$\mathcal{L}_{T \to I} = -\frac{1}{|B|} \sum_{k=1}^{|B|} \log \frac{\exp(z_{kk}^{(S)})}{\sum_{j=1}^{|B|} \exp(z_{jk}^{(S)})} \tag{42}$$

$z_{jk}^{(S)}$ represents the similarity between the $j$-th image embedding and the $k$-th text embedding in the batch for the student model, and $z_{kk}^{(S)}$ is the same as that in (41).

The total contrastive loss, which balances both image-to-text and text-to-image objectives, is computed as:

$$\mathcal{L}_{\text{contrastive}} = \frac{1}{2}(\mathcal{L}_{I \to T} + \mathcal{L}_{T \to I}). \tag{43}$$

This loss function encourages the student model to align its multi-modal representations by bringing matching pairs closer in the embedding space while pushing apart non-matching pairs. Contrastive loss has been extensively applied to knowledge distillation (Chen et al., 2021)(Gao et al., 2021)(Peng et al., 2022)(Zhu et al., 2021)(Guo et al., 2023).

To enhance the effectiveness of distillation, we extend this contrastive loss with additional terms such as KL divergence and transfer entropy-based regularization. These terms further refine the student model's learning dynamics by ensuring information flow from the teacher's embeddings to the student's representations while preserving structural consistency across modalities.

### F.3 KL Divergence for VLM Distillation

To ensure that the student model effectively mimics the probability distributions of the teacher model, we include a Kullback-Leibler (KL) divergence loss term. KL divergence measures how much the student's predicted distribution deviates from the teacher's distribution, enforcing a closer alignment between their logits. KL divergence has been applied to VLM distillation (Li et al., 2024b)(Sun et al., 2024).

For a given batch of image-text pairs, let $\mathbf{z}^{(S)}$ and $\mathbf{z}^{(T)}$ represent the similarity logits of the student and teacher models, respectively. The soft probability distributions are obtained via the softmax function:

$$P_i^{(S)} = \frac{\exp(z_i^{(S)}/\tau)}{\sum_{j=1}^{|B|} \exp(z_j^{(S)}/\tau)}, \tag{44}$$

$$P_i^{(T)} = \frac{\exp(z_i^{(T)}/\tau)}{\sum_{j=1}^{|B|} \exp(z_j^{(T)}/\tau)}, \tag{45}$$

where $\tau$ is the temperature parameter that controls the sharpness of the distributions.

The KL divergence loss is computed as:

$$\mathcal{L}_{\text{KL}} = \frac{1}{2} \left( D_{\text{KL}} \left( P_{\text{image}}^{(S)} \parallel P_{\text{image}}^{(T)} \right) + D_{\text{KL}} \left( P_{\text{text}}^{(S)} \parallel P_{\text{text}}^{(T)} \right) \right), \tag{46}$$

where the KL divergence between two probability distributions $P^{(S)}$ and $P^{(T)}$ is defined as:

$$D_{\text{KL}}(P^{(S)} \parallel P^{(T)}) = \sum_{i=1}^{|B|} P_i^{(T)} \log \frac{P_i^{(T)}}{P_i^{(S)}}. \tag{47}$$

This loss encourages the student model to produce probability distributions that closely resemble those of the teacher, effectively preserving the knowledge distilled from the teacher while allowing the student to generalize efficiently.

### F.4  MSE Loss Function for VLM Distillation

To further align the feature representations of the teacher and student models, we include MSE loss that minimizes the discrepancy between their intermediate embeddings (Yang et al., 2024). The MSE loss is computed as the sum of the squared differences between the student and teacher embeddings for both modalities:

$$\mathcal{L}_{\text{MSE}} = \mathcal{L}_{\text{MSE}}^{\text{image}} + \mathcal{L}_{\text{MSE}}^{\text{text}}, \tag{48}$$

where

$$\mathcal{L}_{\text{MSE}}^{\text{image}} = \frac{1}{|B|} \sum_{i=1}^{|B|} \left\| \hat{\mathbf{v}}_i^{(S)} - \hat{\mathbf{v}}_i^{(T)} \right\|^2, \tag{49}$$

$$\mathcal{L}_{\text{MSE}}^{\text{text}} = \frac{1}{|B|} \sum_{i=1}^{|B|} \left\| \hat{\mathbf{u}}_i^{(S)} - \hat{\mathbf{u}}_i^{(T)} \right\|^2. \tag{50}$$

Here, $|B|$ represents the batch size, and $\| \cdot \|^2$ denotes the squared Euclidean norm. This loss ensures that the student model's learned embeddings remain close to the teacher's representations in the feature space, facilitating effective knowledge transfer. MSE has been applied to VLM loss function in (Yang et al., 2024), and was called feature distillation.

### F.5  Interactive Contrastive Learning

Interactive Contrastive Learning (ICL) was proposed in (Yang et al., 2024) to aligns the student model's feature representations with those of the teacher by treating the student embeddings as anchors and contrasting them with the teacher embeddings.

Given a batch of image-text pairs, let $\mathbf{v}_k^{(S)}$ be the image embedding from the student model, and $\{\mathbf{u}_b^{(T)}\}_{b=1}^{|B|}$ denote the contrastive text embeddings from the teacher model. The image-to-text ICL loss is formulated as:

$$\mathcal{L}_{\text{ICL}}^{I \to T} = - \log \frac{\exp(\mathbf{v}_k^{(S)} \cdot \mathbf{u}_k^{(T)}/\tau)}{\sum_{b=1}^{|B|} \exp(\mathbf{v}_k^{(S)} \cdot \mathbf{u}_b^{(T)}/\tau)}, \tag{51}$$

where $\tau$ is the temperature parameter.

Similarly, for a student text embedding $\mathbf{u}_k^{(S)}$ and contrastive image embeddings from the teacher model $\{\mathbf{v}_b^{(T)}\}_{b=1}^{|B|}$, the text-to-image ICL loss is:

$$\mathcal{L}_{\text{ICL}}^{T \to I} = - \log \frac{\exp(\mathbf{u}_k^{(S)} \cdot \mathbf{v}_k^{(T)}/\tau)}{\sum_{b=1}^{|B|} \exp(\mathbf{u}_k^{(S)} \cdot \mathbf{v}_b^{(T)}/\tau)}. \tag{52}$$

The final ICL loss is a combination of the two:

$$\mathcal{L}_{\text{ICL}} = \frac{1}{2} \left( \mathcal{L}_{\text{ICL}}^{I \to T} + \mathcal{L}_{\text{ICL}}^{T \to I} \right). \tag{53}$$

By integrating ICL, the student model effectively learns from the teacher's structured feature space, leading to improved representation learning and knowledge transfer.

### F.6 Mutual-Information (MI) Objective

In addition to the losses above, we also evaluate mutual-information objectives as an alternative teacher-student supervision signal. This baseline helps distinguish improvements due to directed, conditional information-flow regularization from improvements that may arise from standard contrastive representation matching.

**MI baseline loss (CRD-style).** We implement an MI-driven alignment loss following the contrastive mutual-information maximization principle used in Contrastive Representation Distillation (CRD) (Tian et al., 2020). Using the same notation as above, given a batch of $|B|$ image-text pairs, let $\mathbf{v}_k^{(S)}$ and $\mathbf{u}_k^{(S)}$ denote the student image and text embeddings, and $\mathbf{v}_k^{(T)}$ and $\mathbf{u}_k^{(T)}$ the corresponding teacher embeddings. We $\ell_2$-normalize the embeddings, yielding $\hat{\mathbf{v}}_k^{(\cdot)}$ and $\hat{\mathbf{u}}_k^{(\cdot)}$, and use temperature $\tau$.

Unlike the CLIP-style contrastive loss in Eq. (41) and ICL (Yang et al., 2024), which both involve *cross-modal* matching (image↔text) either within the student or across teacher-student, this MI baseline performs *modality-matched* teacher-student alignment (image↔image and text↔text). Concretely, the image-side MI loss treats $(\hat{\mathbf{v}}_k^{(T)}, \hat{\mathbf{v}}_k^{(S)})$ as the positive pair and $(\hat{\mathbf{v}}_k^{(T)}, \hat{\mathbf{v}}_b^{(S)})$ for $b \neq k$ as negatives:

$$\mathcal{L}_{\mathrm{MI}}^V = -\frac{1}{|B|} \sum_{k=1}^{|B|} \log \frac{\exp\left(\hat{\mathbf{v}}_k^{(T)} \cdot \hat{\mathbf{v}}_k^{(S)}/\tau\right)}{\sum_{b=1}^{|B|} \exp\left(\hat{\mathbf{v}}_k^{(T)} \cdot \hat{\mathbf{v}}_b^{(S)}/\tau\right)}. \tag{54}$$

Similarly, the text-side MI loss treats $(\hat{\mathbf{u}}_k^{(T)}, \hat{\mathbf{u}}_k^{(S)})$ as the positive pair:

$$\mathcal{L}_{\mathrm{MI}}^U = -\frac{1}{|B|} \sum_{k=1}^{|B|} \log \frac{\exp\left(\hat{\mathbf{u}}_k^{(T)} \cdot \hat{\mathbf{u}}_k^{(S)}/\tau\right)}{\sum_{b=1}^{|B|} \exp\left(\hat{\mathbf{u}}_k^{(T)} \cdot \hat{\mathbf{u}}_b^{(S)}/\tau\right)}. \tag{55}$$

The final MI objective is the symmetric combination

$$\mathcal{L}_{\mathrm{MI}} = \tfrac{1}{2} \left( \mathcal{L}_{\mathrm{MI}}^V + \mathcal{L}_{\mathrm{MI}}^U \right). \tag{56}$$

Minimizing $\mathcal{L}_{\mathrm{MI}}$ encourages matched teacher-student representations to score higher than mismatched pairs within the batch, which corresponds to maximizing an InfoNCE lower bound on mutual information (Tian et al., 2020).

### F.7 MSE-Difference Objective

We also evaluate simple $\ell_2$ objectives in place of cosine similarity to test whether improvements can be explained by standard Euclidean change matching rather than directional similarity.

**MSE-difference baseline loss.** To mirror our change-based signals, we compare temporal differences in embeddings (across steps, layers, or time indices depending on the method):

$$\Delta\mathbf{v}_k^{(T)} = \mathbf{v}_k^{(T),(l)} - \mathbf{v}_k^{(T),(l-1)}, \qquad \Delta\mathbf{v}_k^{(S)} = \mathbf{v}_k^{(S),(l)} - \mathbf{v}_k^{(S),(l-1)}, \tag{57}$$

(and analogously for text embeddings $\Delta\mathbf{u}_k^{(T)}, \Delta\mathbf{u}_k^{(S)}$), and define

$$\mathcal{L}_{\mathrm{MSE}\text{-}\Delta} = \frac{1}{2}\left( \mathbb{E}_k\left[\|\Delta\mathbf{v}_k^{(T)} - \Delta\mathbf{v}_k^{(S)}\|_2^2\right] + \mathbb{E}_k\left[\|\Delta\mathbf{u}_k^{(T)} - \Delta\mathbf{u}_k^{(S)}\|_2^2\right] \right). \tag{58}$$

This baseline isolates whether gains arise from a directed information-flow formulation versus simply matching first-order changes under an $\ell_2$ metric.

## G Brief Introduction to the Teacher CLIP Models

We briefly summarize the CLIP architectures used in our experiments.

**CLIP ResNet-50 (RN50).** CLIP RN50 adopts a ResNet-50 backbone as the visual encoder and a Transformer-based text encoder (Radford et al., 2021). The image encoder processes $224 \times 224$ RGB inputs and produces a 1024-dimensional global image embedding, while the text encoder is a 12-layer Transformer (Vaswani et al., 2017) with hidden dimension 512. In the original CLIP implementation, the image encoder has roughly 38M parameters and the text encoder about 63M parameters, for a total of approximately 102M parameters (Radford et al., 2021). In our experiments, CLIP RN50 serves as a moderately sized vision–language teacher model that is well suited for distillation into smaller students.

**CLIP ResNet-50×16 (RN50×16).** RN50×16 is a higher-capacity variant of CLIP's ResNet family (Radford et al., 2021). It scales up RN50 using a ResNeXt-style design with $16\times$ wider convolutional blocks, substantially increasing the number of visual parameters and intermediate feature capacity. The text encoder architecture and the contrastive training objective remain unchanged, but the larger visual backbone yields stronger zero-shot performance and a richer multimodal embedding space compared to RN50.

**CLIP ViT-B/16.** CLIP ViT-B/16 replaces the convolutional backbone with a Vision Transformer (ViT-B/16) (Dosovitskiy et al., 2020; Radford et al., 2021). Images are divided into non-overlapping $16 \times 16$ patches, which are linearly projected and fed into a 12-layer Transformer encoder with hidden dimension 768, producing a global CLS-style image embedding. The text side uses the same 12-layer Transformer encoder with hidden dimension 512 as in RN50. The resulting dual-encoder model has approximately 151M parameters (about 86M in the visual encoder and 63M in the text encoder), and typically achieves stronger zero-shot performance than RN50 due to improved global reasoning and larger effective receptive fields.

**Shared CLIP training paradigm.** Across all three architectures, CLIP jointly trains the vision and text encoders with a contrastive image–text objective so that paired image–text embeddings are close in cosine-similarity space while unpaired pairs are pushed apart (Radford et al., 2021). This unified training scheme enables zero-shot recognition by comparing image embeddings to text-encoded label prompts at inference time, and provides a natural teacher for distilling smaller vision–language students.

## H   Additional Experimental Results

### H.1   Teacher: RN50, Student: RN34, Dataset: Flick8k

We further evaluate our approach on the Flickr8k dataset (Marco et al., 2023), using 85% of the data for training and 15% for testing. Performance results for various loss functions are summarized in Table 11. The loss function employs weighting factors $\alpha = 1.0$, $\beta = 100$, $\delta = 1.0$, $\gamma = 5.0$, and a temperature parameter $\tau = 0.07$. These parameters were selected based on the relative contribution of each loss term to the total loss during training, ensuring balanced optimization. Given the modest size of Flickr8k, all experiments were conducted on a Google Colab instance equipped with an A100 GPU and limited system RAM. Each experiment (i.e., each row in Table 11) required approximately 20 minutes of training time. Notably, incorporating TE1 or TE2 into the loss function consistently improves both image-to-text (I2T) and text-to-image (T2I) retrieval performance compared to baselines that rely solely on standard distillation losses such as CL + KL or CL + MSE. These results underscore the effectiveness of transfer entropy approximations in guiding student model updates during distillation.

Table 12 reports the hyperparameter sensitivity of zero-shot retrieval when distilling a student RN34 from an RN50 teacher on Flickr8k. Overall, incorporating the TE reward has a clear and consistent impact: relative to the baseline without TE ($\gamma = 0$), increasing $\gamma$ improves both I2T and T2I performance, with the best overall results achieved at a moderate TE weight. In particular, $\gamma = 5$ yields the strongest bidirectional retrieval, attaining the highest I2T R@5/R@10 (65.65/77.84) and the best T2I R@1/R@5/R@10 (27.00/57.20/70.43), while slightly larger $\gamma$ (e.g., 7.5) maintains similar I2T R@1 but reduces T2I, indicating diminishing returns and mild over-regularization at high TE weights. Adjusting the other loss weights has comparatively smaller effects: increasing $\alpha$ to 5 or $\beta$ to 100 under $\gamma = 5$ produces performance close to, but not exceeding, the best setting, whereas increasing the ICL weight ($\delta = 5$) noticeably degrades retrieval, suggesting that overly strong interaction-level supervision can be harmful on this smaller dataset. Adding the MI term

Table 11: Zero-shot retrieval performance (Recall@k) on **Flickr8k** of RN34-based VLM student using teacher CLIP RN50 under different loss functions. All Loss Function: CL + KL + MSE + ICL - TE1 - TE2. The weighting factors for loss functions: $\alpha = 1.0$, $\beta = 100$, $\delta = 1.0$, $\gamma = 5.0$.

| Model and Loss Function | I2T Retrieval (R) | | | T2I Retrieval (R) | | |
|---|---|---|---|---|---|---|
| | R@1 | R@5 | R@10 | R@1 | R@5 | R@10 |
| **Teacher Model (RN50)** | 51.65% | 78.17% | 87.73% | 47.28% | 75.21% | 84.60% |
| **Student Models (RN34)** | | | | | | |
| CL Only (Oord et al., 2018) | 22.73% | 48.19% | 60.87% | 18.47% | 43.76% | 56.77% |
| CL + MSE (Yang et al., 2024) | 22.98% | 49.92% | 62.52% | 17.84% | 44.71% | 57.99% |
| CL + KL (Li et al., 2024b) | 27.51% | 56.51% | 69.19% | 23.20% | 50.12% | 62.82% |
| CL + ICL (Yang et al., 2024) | 24.55% | 52.06% | 64.50% | 19.87% | 47.97% | 61.24% |
| CL - TE1 | 30.48% | 62.52% | 74.05% | 24.42% | 54.25% | 68.39% |
| CL - TE2 | 31.80% | 61.37% | 72.90% | **25.19%** | 54.66% | **68.70**% |
| CL - TE1 - TE2 | 32.29% | 62.36% | **75.29**% | 24.50% | 54.56% | 68.14% |
| CL + KL + MSE - TE1 | 31.22% | 59.31% | 71.99% | 23.64% | 53.20% | 67.27% |
| CL + KL + ICL - TE1 | 31.22% | 59.88% | 72.08% | 24.66% | 53.05% | 66.34% |
| CL + KL + MSE + ICL - TE1 | 30.23% | 58.81% | 72.24% | 24.18% | 53.16% | 66.64% |
| All Loss Function | **34.76%** | **63.43%** | 74.14% | 24.50% | **55.14%** | 68.29% |

alone ($\eta \in \{1, 2.5, 5, 7.5\}$ with $\gamma = 0$) yields improvements over the baseline and can approach the best TE-regularized performance (e.g., $\eta = 5$ achieves I2T R@10 of 77.10 and T2I R@5 of 57.03), but combining MI and MSE-$\Delta$ with TE does not provide consistent additive gains within this sweep. Taken together, these results indicate that Flickr8k benefits most from moderate TE regularization (around $\gamma = 5$), while the overall objective remains relatively robust to moderate variations in the remaining hyperparameters.

Table 12: Comparison of zero-shot retrieval performance (Recall@k) of RN34-based VLM student with teacher CLIP RN50 on Flick8k.

| $\alpha$ | $\beta$ | $\delta$ | $\eta$ | $\lambda$ | $\gamma$ | I2T R@1 | I2T R@5 | I2T R@10 | T2I R@1 | T2I R@5 | T2I R@10 |
|---|---|---|---|---|---|---|---|---|---|---|---|
| 1 | 50 | 1 | 0 | 0 | 0 | 26.61% | 58.15% | 69.93% | 22.49% | 51.48% | 64.74% |
| 1 | 50 | 1 | 0 | 0 | 1 | 31.55% | 61.70% | 71.99% | 24.38% | 53.86% | 67.84% |
| 1 | 50 | 1 | 0 | 0 | 2.5 | 35.34% | 61.94% | 73.72% | 25.62% | 53.11% | 67.17% |
| 1 | 50 | 1 | 0 | 0 | 5 | 35.42% | **65.65%** | **77.84%** | **27.00%** | **57.20%** | **70.43%** |
| 1 | 50 | 1 | 0 | 0 | 7.5 | **35.50%** | 63.67% | 75.45% | 25.55% | 55.42% | 68.62% |
| 1 | 50 | 1 | 0 | 0 | 10 | 33.20% | 62.27% | 73.72% | 23.90% | 54.94% | 68.95% |
| 5 | 50 | 1 | 0 | 0 | 5 | 34.10% | 63.26% | 75.29% | 25.17% | 54.00% | 68.27% |
| 1 | 100 | 1 | 0 | 0 | 5 | 33.53% | 64.50% | 75.86% | 25.67% | 55.30% | 69.79% |
| 1 | 50 | 5 | 0 | 0 | 5 | 29.65% | 59.47% | 73.72% | 24.32% | 52.39% | 65.98% |
| 1 | 50 | 1 | 0 | 50 | 0 | 29.00% | 58.48% | 70.10% | 23.25% | 51.66% | 65.55% |
| 1 | 50 | 1 | 0 | 100 | 0 | 29.08% | 58.24% | 70.68% | 24.18% | 52.83% | 66.39% |
| 1 | 50 | 1 | 1 | 1 | 0 | 32.70% | 62.60% | 74.14% | 26.00% | 54.71% | 67.64% |
| 1 | 50 | 1 | 2.5 | 0 | 0 | 31.30% | 60.96% | 73.39% | 24.18% | 53.33% | 67.23% |
| 1 | 50 | 1 | 5 | 0 | 0 | 34.02% | 65.16% | 77.10% | 26.24% | 57.03% | 69.67% |
| 1 | 50 | 1 | 7.5 | 0 | 0 | 33.20% | 64.66% | 75.95% | 25.27% | 53.99% | 67.78% |
| 1 | 50 | 1 | 2.5 | 50 | 2.5 | 32.13% | 62.77% | 75.37% | 25.02% | 53.99% | 67.51% |
| 1 | 50 | 1 | 2.5 | 0 | 5 | 31.14% | 64.99% | 76.28% | 24.68% | 55.90% | 69.09% |
| 1 | 50 | 1 | 2.5 | 50 | 5 | 33.61% | 62.19% | 74.79% | 24.84% | 55.58% | 68.86% |
| 1 | 50 | 1 | 1 | 0 | 7.5 | 31.14% | 62.44% | 74.63% | 24.81% | 55.14% | 68.55% |
| 1 | 50 | 1 | 1 | 50 | 7.5 | 32.95% | 62.44% | 74.14% | 26.33% | 55.67% | 68.65% |
| 1 | 50 | 1 | 5 | 50 | 7.5 | 30.15% | 60.87% | 74.55% | 23.84% | 54.55% | 68.01% |

## H.2 Teacher: RN50, Student Model: RN18

In addition to using RN34 as the student model, we also conduct experiments with RN18 as the student image encoder. The RN18 architecture is a more compact variant, containing approximately 11.7 million parameters (He et al., 2016). Similar to RN34, the final fully connected layer is modified to output 1024-dimensional features, keeping the overall parameter count stable. Given that the text encoder remains unchanged, the total number of parameters for the RN18-based student model is approximately 45-50 million. This reduction in model size compared to the RN34-based student allows for a more lightweight design while still leveraging the benefits of contrastive learning and effective knowledge transfer from the teacher model.

We applied the same loss components and hyperparameter settings as in Section 5: $\alpha = 1.0$, $\beta = 50$, $\delta = 1.0$, $\gamma = 1.0$, and a temperature parameter $\tau = 0.05$. Figure 4 presents the training losses and TE rewards over epochs for various configurations. Compared to RN34, RN18 exhibits a similar trend where the total training loss steadily decreases, and TE rewards increase over epochs, indicating effective optimization and knowledge transfer. However, due to the smaller capacity of RN18, the absolute TE rewards remain slightly lower than those observed for RN34, suggesting a less expressive feature alignment between teacher and student. Furthermore, the KL loss and MSE components show even less significant reductions over training epochs, likely due to the more limited representational capacity of RN18. This highlights that while TE-based regularization remains effective in guiding knowledge distillation, the overall learning dynamics are constrained by the smaller network size, making RN34 a more effective student model in terms of retaining structured alignment with the teacher.

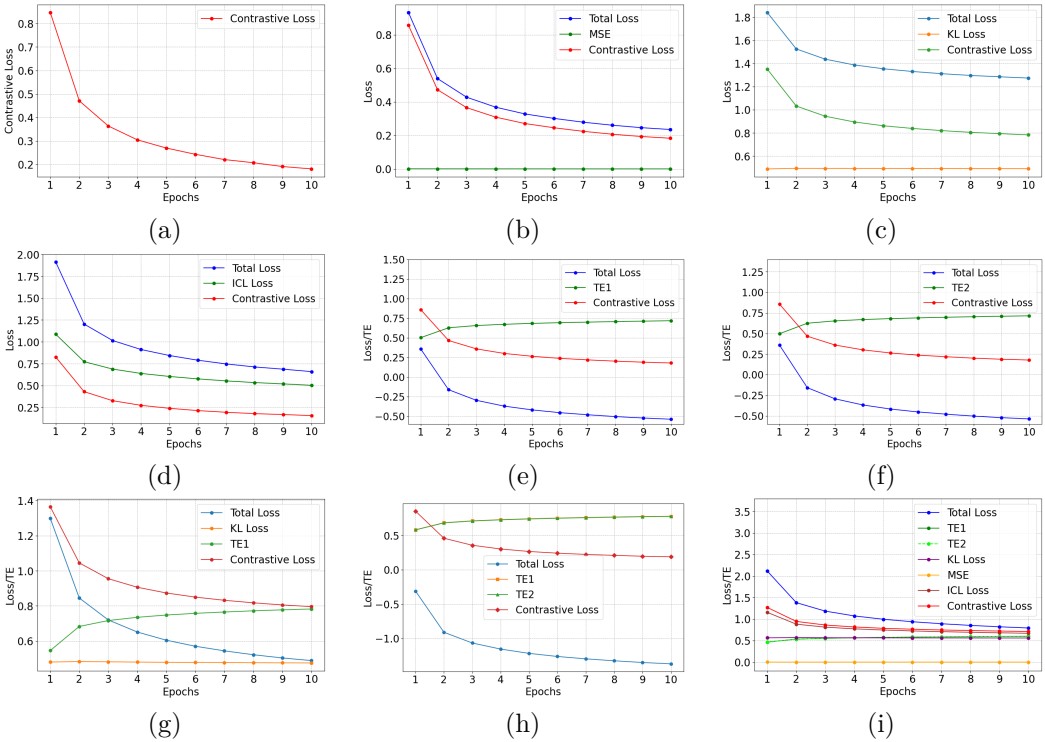

Figure 4: The training losses and TE for different loss functions in the training of RN18-based VLM Student with teacher CLIP RN50. (a) Contrastive only, (b) Contrastive + MSE, (c) Contrastive + KL, (d) Contrastive + ICL, (e) Contrastive - TE1, (f) Contrastive - TE2, (g) Contrastive + KL - TE1, (h) Contrastive - TE1 - TE2, (i) Contrastive + KL + ICL + MSE - TE1 - TE2.

We used Google Colab Pro with a T4 GPU and High-RAM for training and evaluating RN18. Due to its significantly fewer parameters compared to RN34, the student model RN18 required less training time. We trained it for 10 epochs in each loss function combination scenario, with the training and evaluation process taking approximately 11 hours per experimental setup.

We summarize the zero-shot retrieval performance for the trained RN18 student model in Table 13. Similar observations we can make that the experiment with loss function (Contrastive - TE1 -TE2) achieved the best performance for Image-to-Text retrieval, while the experiment with loss function (Contrastive + KL + MSE + ICL - TE1 - TE2) achieved the best performance in Text-to-Image Retrieval. Comparing Table 13 with Table 1, the results indicate that while RN18 achieves competitive performance across different loss function combinations, it underperforms compared to RN34 for all loss configurations, with RN34 consistently yielding higher Recall@k values. However, the best-performing RN18 model (Contrastive - TE1 - TE2) achieves Recall@1 of 6.65% for image-to-text retrieval, which is not far behind RN34's highest Recall@1 values of 8.24% under the same loss formulation. This suggests that while RN18 is a lighter-weight alternative, RN34 remains a better choice for preserving retrieval performance during distillation. The trade-off between model complexity and retrieval accuracy highlights the importance of selecting an appropriate student architecture based on deployment constraints and performance requirements.

Table 13: Comparison of zero-shot retrieval performance (Recall@k) of RN18-based VLM student with teacher CLIP RN50 for different loss function combinations in VLM distillation using MSCOCO. All Loss Function: CL + KL + MSE + ICL - TE1 - TE2.

| Model and Loss Function | I2T Retrieval (R) | | | T2I Retrieval (R) | | |
|---|---|---|---|---|---|---|
| | R@1 | R@5 | R@10 | R@1 | R@5 | R@10 |
| **Teacher Model (RN50)** | 15.27% | 30.73% | 39.05% | 11.68% | 25.52% | 33.50% |
| **Student Models (RN18)** | | | | | | |
| CL Only (Oord et al., 2018) | 4.38% | 13.28% | 20.40% | 3.39% | 11.07% | 17.22% |
| CL + MSE (Yang et al., 2024) | 4.27% | 13.29% | 20.15% | 3.47% | 11.17% | 17.28% |
| CL + KL (Li et al., 2024b) | 4.89% | 15.23% | 22.90% | 4.58% | 13.99% | 21.05% |
| CL + ICL (Yang et al., 2024) | 5.39% | 15.48% | 22.95% | 4.32% | 13.23% | 19.96% |
| CL - TE1 | 5.48% | 16.43% | 24.59% | 4.60% | 13.86% | 20.80% |
| CL - TE2 | 5.57% | 16.67% | 24.78% | 4.67% | 14.08% | 20.97% |
| CL - TE1 - TE2 | **6.65%** | **18.75%** | **27.33%** | 5.18% | 15.09% | 22.35% |
| CL + KL - TE1 | 6.49% | 18.37% | 26.83% | 5.18% | 15.17% | 22.52% |
| All Loss Function | 6.52% | 18.60% | 27.16% | **5.79%** | **16.78%** | **24.47%** |

### H.3 Teacher: RN50, Student: RN34, Application in Classification

We have evaluated our TE-based distillation on Food-101 (Bossard et al., 2014), a challenging benchmark dataset for large-scale food recognition. Food-101 contains 101 categories with a total of 101,000 images, split into 75,750 images for training and 25,250 images for testing. This dataset is particularly suitable for evaluating knowledge transfer since it combines significant intra-class variation with a large number of categories, which makes direct zero-shot transfer difficult for a smaller-capacity student network.

In our setup, the teacher is a ResNet-50 (RN50) model, and the student is a smaller ResNet-34 (RN34). Importantly, during distillation, the student is trained without direct access to the ground truth labels. Instead, it learns only from the outputs of the teacher, thereby relying entirely on the transferred information. This design allows us to directly measure the effectiveness of the proposed TE-based framework in capturing and transferring generalizable knowledge from teacher to student.

Table 14 summarizes the zero-shot classification accuracy of the student RN34 under different weightings of the loss components (cf. Eq. 19), alongside the teacher RN50 baseline. Several key observations emerge. First, the naive baseline where $\gamma = 0$ (i.e., without TE) performs better than the teacher in terms of Top-1 accuracy but slightly underperforms in Top-5 accuracy. Second, once TE is introduced ($\gamma > 0$), we observe consistent improvements across both Top-1 and Top-5 accuracy. For instance, setting $\gamma = 2.5$ increases the student's Top-1 accuracy to 82.46% and Top-5 accuracy to 96.23%, surpassing the teacher by significant margins. Larger $\gamma$ values generally sustain these gains, with $\gamma = 7.5$ yielding the best Top-5 performance (96.62%), and an alternative setting with $\alpha = 5$ and $\gamma = 2.5$ providing the overall best Top-1 accuracy

(82.91%). These trends suggest that TE contributes complementary signal during distillation that is not fully captured by conventional loss terms. Each experiment (each row) in Table 14 takes around 45 minutes using Colab with GPU A100.

Table 14: Zero-shot classification accuracy (%) of RN34-based VLM student and teacher CLIP RN50 on Food-101.

| $\alpha$ | $\beta$ | $\delta$ | $\eta$ | $\lambda$ | $\gamma$ | Top-1 Acc. | Top-5 Acc. |
|---|---|---|---|---|---|---|---|
| 1 | 50 | 1 | 0 | 0 | 0 | 80.23% | 95.22% |
| 1 | 50 | 1 | 0 | 0 | 2.5 | 82.46% | 96.23% |
| 1 | 50 | 1 | 0 | 0 | 5 | 82.37% | 96.44% |
| 1 | 50 | 1 | 0 | 0 | 7.5 | 82.27% | **96.62%** |
| 1 | 50 | 1 | 0 | 0 | 10 | 82.01% | 96.38% |
| 5 | 50 | 1 | 0 | 0 | 2.5 | **82.91%** | 96.47% |
| 1 | 100 | 1 | 0 | 0 | 2.5 | 82.54% | 96.10% |
| 1 | 50 | 5 | 0 | 0 | 2.5 | 81.07% | 95.30% |
| 1 | 50 | 1 | 2.5 | 0 | 0 | 82.13% | 96.10% |
| 1 | 50 | 1 | 5 | 0 | 0 | 82.37% | 96.50% |
| 1 | 50 | 1 | 7.5 | 0 | 0 | 81.39% | 96.24% |
| 1 | 50 | 1 | 5 | 50 | 7.5 | 81.63% | 96.29% |
| Teacher (RN50) | | | | | | 79.80% | 96.17% |

Overall, our results demonstrate that the student RN34, despite its smaller capacity, is able to not only match but even surpass the teacher RN50 under several configurations. This improvement cannot be attributed to overfitting, since no ground truth labels are used during distillation, but instead highlights the effectiveness of TE-based distillation in transferring structured, generalizable information. This experiment thus provides strong evidence that TE is a valuable component for enhancing knowledge transfer in classification tasks.

### H.4    Zero-Shot Classification on ImageNet-1k

To further assess how much semantic structure the student retains beyond Food-101, we evaluate zero-shot classification on the standard ImageNet-1k benchmark (Deng et al., 2009)(Russakovsky et al., 2015). We distill an RN34 student from an RN50 teacher using MSCOCO and Food-101, and then perform zero-shot classification on the ImageNet-1k validation set (50K images) following the standard CLIP-style protocol. Importantly, the student is *never* trained on ImageNet images; thus, this experiment measures out-of-domain semantic transfer under a substantial domain shift.

Table 15 reports Top-1 and Top-5 accuracies for representative hyperparameter settings. Overall, incorporating the TE reward yields the strongest ImageNet performance among the tested configurations: using $\gamma \in \{5, 7.5\}$ achieves Top-1 accuracy of 10.23%–10.62% and Top-5 accuracy of 24.08%–24.46%, outperforming MI-only variants (e.g., $\eta \in \{2.5, 5\}$ with $\gamma = 0$), which attain Top-1 accuracy of 8.09%–8.61% and Top-5 accuracy of 19.92%–21.24%. This trend is consistent with our retrieval ablations, where moderate TE regularization improves alignment and yields better generalization. While absolute ImageNet accuracy remains well below the RN50 teacher, as expected given the student's smaller capacity and the fact that distillation is performed only on MSCOCO and Food-101, the improvement from TE indicates that TE-guided distillation helps preserve transferable semantic information under a challenging out-of-domain evaluation.

### H.5    Qualitative Embedding-Space Visualization

This section provides a qualitative view of how the student and teacher organize the CLIP embedding space after 10 epochs of distillation on the MSCOCO training set. We evaluate on the MSCOCO validation split and visualize teacher and student embeddings computed on the *same* held-out samples.

Table 15: Zero-shot classification accuracy (%) of distilled RN34-based VLM student on ImageNet-1k.

| $\alpha$ | $\beta$ | $\delta$ | $\eta$ | $\lambda$ | $\gamma$ | Top-1 Acc. | Top-5 Acc. |
|---|---|---|---|---|---|---|---|
| 1 | 50 | 1 | 0 | 0 | 5 | 10.23% | 24.08% |
| 1 | 50 | 1 | 0 | 0 | 7.5 | 10.62% | 24.46% |
| 1 | 50 | 1 | 2.5 | 0 | 0 | 8.09% | 19.92% |
| 1 | 50 | 1 | 5 | 0 | 0 | 8.61% | 21.24% |

**Teacher-only PCA protocol.** For each modality (image and text), we fit PCA *only on the teacher* validation embeddings and project both teacher and student embeddings using this shared 2D basis (Figs. 5 and 6). Because PCA is fit separately for each experiment (TE vs. MI), absolute PCA coordinates are not directly comparable across figures; we therefore focus on teacher-student *overlap within each figure* and complement the visualization with local-geometry and per-sample agreement metrics.

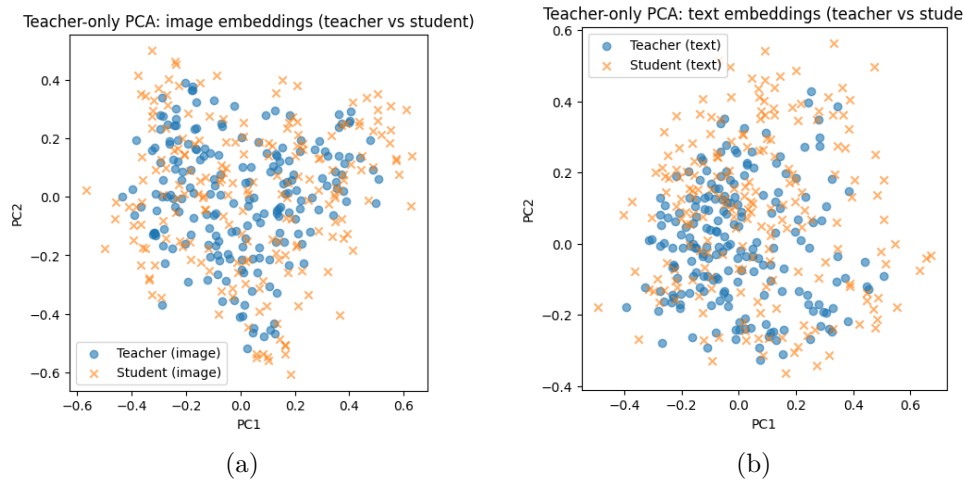

Figure 5: Teacher-only PCA visualization of teacher–student embedding geometry on MSCOCO val after 10 epochs of distillation on MSCOCO train (TE-based; $\gamma = 7.5$, with $\alpha = 1$, $\beta = 50$, $\delta = 1$, $\eta = 0$, $\lambda = 0$). For each modality, PCA is fit on the *teacher* validation embeddings and both teacher and student embeddings are projected using this shared 2D basis. (a) Image embeddings. (b) Text embeddings.

**kNN neighborhood agreement and cosine similarity.** To quantify local-geometry preservation, we compute kNN overlap@10 (after 10 epochs of training) between teacher and student *image* embeddings on MSCOCO val. For each sample $i$, let $\mathcal{N}_{10}^T(i)$ and $\mathcal{N}_{10}^S(i)$ denote its 10 nearest neighbors under the teacher and student embeddings, respectively. We report

$$\frac{1}{N} \sum_{i=1}^{N} \frac{\left| \mathcal{N}_{10}^T(i) \cap \mathcal{N}_{10}^S(i) \right|}{10}, \tag{59}$$

along with the standard deviation across samples. We additionally report per-sample cosine similarity between matched teacher and student embeddings (teacher vs. student) for both modalities.

**TE yields stronger teacher-faithfulness than MI.** Under TE-based distillation ($\gamma = 7.5$), the student more closely matches the teacher distribution in both modality-specific PCA projections (Fig. 5). This is consistent with stronger alignment metrics (Table 16): image-embedding kNN overlap@10 is $0.605 \pm 0.202$, and per-sample cosine similarity is $0.726 \pm 0.078$ for images and $0.840 \pm 0.044$ for text. In contrast, MI-based distillation ($\eta = 5$) exhibits a larger teacher-student shift in both modalities (Fig. 6) and substantially weaker alignment: kNN overlap@10 decreases to $0.546 \pm 0.200$, while cosine similarity drops to $0.487 \pm 0.054$

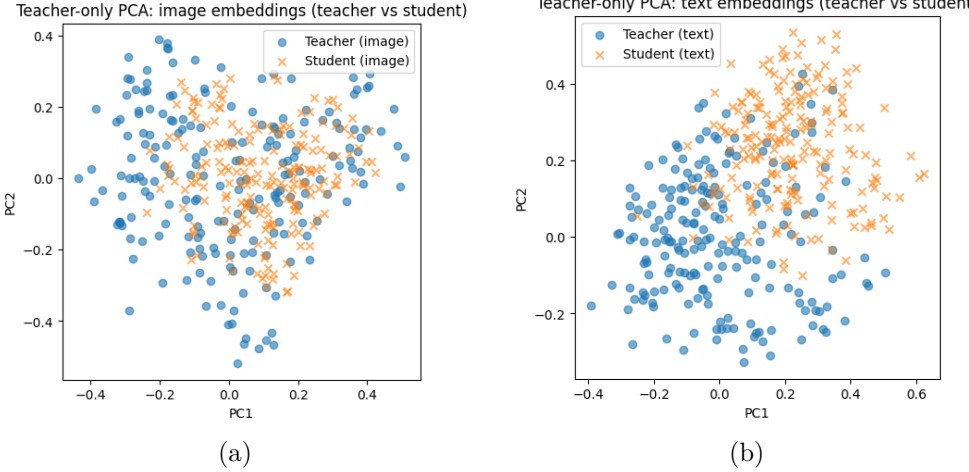

(a)              (b)

Figure 6: Teacher-only PCA visualization of teacher–student embedding geometry on MSCOCO val after 10 epochs of distillation on MSCOCO train (MI-based; $\eta = 5$, with $\alpha = 1$, $\beta = 50$, $\delta = 1$, $\lambda = 0$, $\gamma = 0$). For each modality, PCA is fit on the *teacher* validation embeddings and both teacher and student embeddings are projected using this shared 2D basis. (a) Image embeddings. (b) Text embeddings.

Table 16: Retrieval and embedding-alignment metrics on MSCOCO val after 10 epochs of distillation on MSCOCO train (RN50 teacher, RN34 student). Retrieval metrics are in %, higher is better. kNN overlap@10 measures local neighborhood agreement between teacher and student image embeddings; cosine similarities are per-sample teacher–student agreement in each modality.

| Method | Image→Text | | | | Text→Image | | | |
|---|---|---|---|---|---|---|---|---|
| | R@1 | R@5 | R@10 | MRR | R@1 | R@5 | R@10 | MRR |
| TE ($\gamma = 7.5$) | 10.28 | 25.88 | 35.83 | 18.66 | 6.88 | 18.94 | 27.07 | 13.67 |
| MI ($\eta = 5$) | 8.74 | 23.19 | 33.18 | 16.71 | 7.62 | 20.65 | 29.16 | 14.82 |

| Method | kNN overlap@10 (img) | | $\cos(T_{\text{img}}, S_{\text{img}})$ | | $\cos(T_{\text{txt}}, S_{\text{txt}})$ | |
|---|---|---|---|---|---|---|
| | mean | std | mean | std | mean | std |
| TE ($\gamma = 7.5$) | 0.605 | 0.202 | 0.726 | 0.078 | 0.840 | 0.044 |
| MI ($\eta = 5$) | 0.546 | 0.200 | 0.487 | 0.054 | 0.528 | 0.048 |

(images) and $0.528 \pm 0.048$ (text). Overall, TE better preserves both per-sample correspondence and local neighborhood structure.

**Joint (image+text) structure.** Figure 7 visualizes the *joint* (image+text) embedding geometry by pooling image and text embeddings and projecting them to 2D with teacher-only PCA (fit separately per panel); circles denote image embeddings and crosses denote text embeddings. The teacher (Fig. 7a) exhibits a clear bimodal layout in which image and text clusters are well separated. The TE-distilled student (Fig. 7b) largely preserves this separation, maintaining a teacher-like global organization. In contrast, the MI-distilled student (Fig. 7c) shows substantially more cross-modal mixing and reduced separation, indicating a departure from the teacher's bimodal joint-space structure. As with the per-modality visualizations, we interpret relative separation and overlap *within* each panel rather than comparing absolute coordinates across panels.

**Connection to retrieval.** Table 16 summarizes retrieval and embedding-alignment metrics on MSCOCO val. TE improves Image→Text retrieval relative to MI (R@1 10.28% vs. 8.74%, R@5 25.88% vs. 23.19%, R@10 35.83% vs. 33.18%, MRR 18.66% vs. 16.71%). MI is slightly stronger on Text→Image (R@1 7.62%

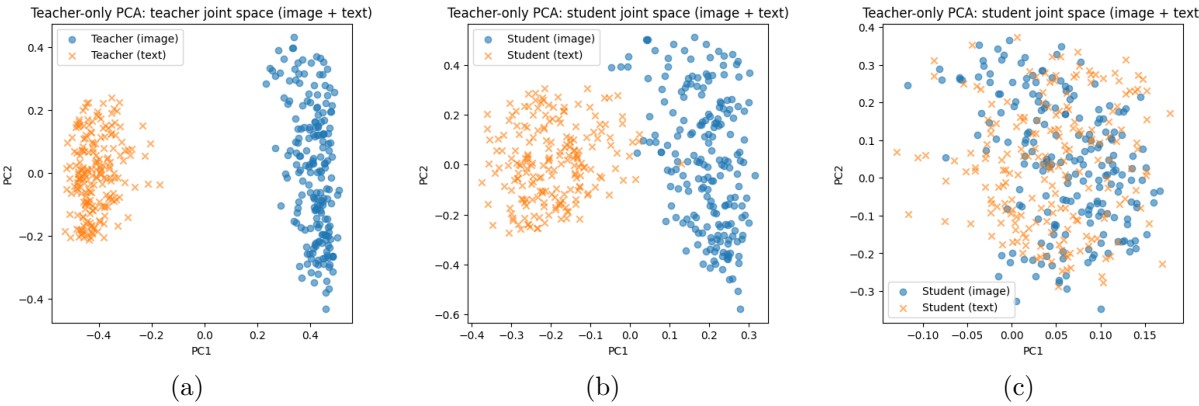

Figure 7: Joint (image+text) embedding visualization on MSCOCO val after 10 epochs of distillation on MSCOCO train. We pool image (circles) and text (crosses) embeddings and project them to 2D with teacher-only PCA (fit separately per panel). (a) Teacher CLIP RN50. (b) TE-distilled RN34 student ($\gamma = 7.5$; $\alpha = 1$, $\beta = 50$, $\delta = 1$, $\eta = 0$, $\lambda = 0$). (c) MI-distilled RN34 student ($\eta = 5$; $\alpha = 1$, $\beta = 50$, $\delta = 1$, $\gamma = 0$, $\lambda = 0$).

vs. 6.88%, R@5 20.65% vs. 18.94%, R@10 29.16% vs. 27.07%, MRR 14.82% vs. 13.67%), suggesting that improved teacher-faithfulness does not uniformly translate to gains in both retrieval directions.

# I Statement of Broader Impact

The development of efficient knowledge distillation frameworks for VLMs, such as our proposed TE-VLM with transfer entropy regularization, has the potential to produce several positive societal impacts. By enabling the deployment of high-performing, compact models on resource-constrained devices, this work can broaden access to advanced AI technologies for individuals and communities with limited computational resources. This democratization may facilitate wider adoption in applications such as assistive technologies for people with disabilities, low-cost language translation tools, and educational platforms. Additionally, parameter-efficient VLMs reduce energy consumption and the environmental footprint associated with large-scale model training and inference, supporting the development of more sustainable and eco-friendly AI systems.

However, the widespread deployment of efficient VLMs also raises potential negative societal impacts. Greater accessibility to vision-language technology may amplify risks related to privacy, surveillance, and misuse, including unauthorized content analysis or the automated generation of misleading media. Furthermore, model distillation may inadvertently propagate or amplify biases present in the teacher model, potentially resulting in unfair or discriminatory outcomes for underrepresented groups. The process of distillation and embedding compression may also lead to the loss of nuanced information, degrading model fairness or accuracy in real-world scenarios.

To alleviate these negative impacts, we advocate for the following measures: (1) incorporating bias and fairness audits throughout the model development and distillation process, particularly for sensitive or high-impact applications; (2) implementing robust data governance and privacy-preserving mechanisms when deploying VLMs in real-world settings; (3) maintaining transparency by publishing model cards, evaluation results, and details of the distillation pipeline; and (4) encouraging interdisciplinary collaboration with ethicists, domain experts, and impacted stakeholders to continuously assess societal risks. By proactively addressing these concerns, the benefits of TE-based VLM distillation can be realized more safely and equitably.

