# OpenReview forum: "TE-VLM: Transfer Entropy for Vision Language Model Distillation"
_TMLR — Rejected by TMLR_

### Review · Reviewer_az7q · 2025-12-06

**Summary Of Contributions:**

This paper introduces a information-theoretic framework for Vision Language Model distillation, centered on the use of Transfer Entropy to explicitly model and maximize the directional flow of knowledge from a large teacher model to a compact student model. Unlike traditional distillation methods which focus primarily on aligning embeddings, logits, or contrastive similarity, this work recognizes that effective knowledge transfer also requires aligning how representations evolve during training, not just their final forms.

**Audience:**

Yes

**Audience Explanation:**

The paper discusses the vllm distillation and a method via information theory. Both the task and the methodology are interesting to the audience of TMLR.

**Broader Impact Concerns:**

No Broader Impact Concerns

**Claims And Evidence:**

Yes

**Claims Explanation:**

Not always, but the authors generally provides clear and convincing evidence supporting its claims.

1. The experiments consistently show that TE-based regularization improves retrieval performance across multiple teacher–student settings and datasets, and the ablation studies validate the specific contribution of TE.

2. Theoretical analysis and empirical trends further reinforce the soundness and accuracy of the proposed approach.

Some issues will be discussed in the later Weakness section.

**Requested Changes:**

Major Issues

1. The entire experimental section effectively functions as an ablation study across multiple benchmarks but lacks comparisons against meaningful alternative baselines. For instance, variants based on mutual information, entropy-based objectives, or simple MSE differences (instead of cosine similarity) are not included. Without such comparisons, it is difficult for readers to understand why the proposed TE-based loss is preferable over other plausible options.

2. The proposed approach appears to modify the standard training workflow by requiring the loss to be computed twice on the same batch. If this interpretation is correct, the method is not easily implementable nor readily pluggable into existing distillation pipelines, raising concerns about practicality and scalability.

3. The teacher models used in the experiments are relatively modest in capacity. It remains unclear whether the proposed TE-based distillation would still provide improvements when applied to stronger, more competitive teacher models. A study with more capable teachers would help assess the generality and robustness of the method.

Minor issues:

1. I encourage the authors to avoid using S to denote both the student model and the text embedding. Its meaning shifts depending on the context or superscript, which can be confusing for readers. Using distinct symbols for these two concepts would greatly improve clarity.

2. I recommend avoiding the use of italic TE in some places and upright TE in Equation (18). For consistency with other loss components, it should always be written as L_\text{TE}. Though it works as a "reward" instead of loss, its reversion could be a loss.

3. The ordering of legends and colors in Figure 1 should be consistent across subplots. In particular, the green and orange curves swap meanings between Panels (d) and (e), causing confusion about which line represents TE and which represents contrastive loss.

4. Some best values are not emphasized properly, such as "20.55% 29.03%" for the second-last row in Table 2.

---

> ### Author Response · Authors · 2026-01-05
> **Stronger Baselines, Practical Workflow, and More Competitive Teachers**
>
> We sincerely thank the reviewers for their constructive and insightful comments. We have revised the paper accordingly.
>
> **Requested Changes:**
>
> **Major 1** We agree that comparisons to plausible alternative objectives help clarify the benefit of our TE-based loss. In the revision, we have added a mutual-information (MI) baseline using the contrastive MI objective from Contrastive Representation Distillation (Tian et al., ICLR’20), in Appendix F.6 and Section 4.3. We report results side-by-side with our TE loss across the same benchmarks (in Tables 2, 4, 6, 8, 9, 10, 12, 14, 15 for $\eta >0$ cases ). In addition, because the review notes “MSE differences instead of cosine similarity,” we have included an ablation that replaces cosine-based alignment with an MSE/L2 objective while keeping the rest of the pipeline unchanged in Appendix F.7, Section 4.3, and Tables 2, 4, 6, 12 ($\lambda>0$ cases). This isolates the effect of the similarity choice from the TE signal itself. The MI-based approach is a strong base line and it achieves competitive performance.
>
> **Major 2** Our method does not require computing the full loss twice on the same batch. TE is defined across optimization steps ($t\rightarrow t+1$): we cache teacher features once (teacher is frozen) and use a small feature buffer to store the student features pre-update and reuse the post-update student features from the next step for the same examples. This is a lightweight wrapper around a standard distillation loop and is readily pluggable. We have revised Section 4.1 to clarify the workflow. Importantly, we compute TE using our proposed approximation in Section 4.2, which substantially reduces computation for practical use.
>
> **Major 3** In the revision, we have added Section 5.4 for More Competitive Teacher Model.
>
> **Minor 1** We have used $U$ and $\textbf{u}$ to denote the text embedding in the revision.
>
> **Minor 2** We have updated and used upright TE for all equations.
>
> **Minor 3** We have fixed Figure 1 to make all legend color consistent across subplots.
>
> **Minor 4** We have made sure all best values are emphasized in the Tables.

---

### Review · Reviewer_cCMV · 2025-12-11

**Summary Of Contributions:**

This paper proposes a distillation framework for CLIP-style vision and language models that takes inspiration from transfer entropy (TE). The theoretical section relates TE, under a linear Gaussian approximation, to the cosine similarity between teacher and student Jacobians. Motivated by this, the paper introduces two practical TE surrogates, TE1 and TE2, based on cosine similarities of embedding differences (directions). These surrogates are added to standard distillation losses such as CL, KL, MSE and ICL. The method is evaluated on several teacher and student pairs across MSCOCO and Flickr8k, with additional analyses in the appendix. In all these cases, the TE-based objectives lead to improvements in retrieval performance.

**Audience:**

Yes

**Audience Explanation:**

Yes. Distillation of vision and language models is of broad interest, and even without a fully realized TE interpretation the empirical improvements seem to indicate that the proposed surrogates could be useful in practice. The theoretical angle may also encourage further work on the structure of teacher–student dynamics.

**Broader Impact Concerns:**

The broader impact aligns with that of standard VLM work. The method makes smaller models more accessible and may also propagate dataset biases. A short reflection on how TE-based distillation interacts with domain shift or fairness-sensitive settings could be a useful addition.

**Claims And Evidence:**

No

**Claims Explanation:**

**Clear empirical gains, but the conceptual connection to TE needs sharpening.**
Across the main experiments, adding TE1 or TE2 improves retrieval scores in a stable way. This part of the contribution is convincing. The conceptual transition from transfer entropy to the practical surrogate losses is less clear. Section 4.1 defines TE along optimization steps for the same batch, with t indexing updates. Section 4.2 then switches to defining TE1 and TE2 using differences between consecutive elements in a batch, which assumes that batch ordering approximates temporal ordering. The text earlier also notes that batches are shuffled and that t should be interpreted as the learning step. These different uses of the time dimension pull in opposite directions and make it hard to interpret the surrogates as meaningful approximations of TE as defined in Section 4.1.

**The interpretation of TE shifts throughout the paper.**
The teacher is frozen, yet Sections 4.1 and 4.2.1 refer to directional changes in both teacher and student representations. The claim that TE is high at the beginning (to account for the stronger directionality mismatch) of training contrasts with the observed increasing trend of the TE surrogate over epochs in Figure 1. The surrogates themselves are symmetric cosine similarities, so any directional interpretation is lost. None of this diminishes the empirical usefulness of TE1 and TE2, but it does weaken the theoretical narrative around directional information flow.

**The experiments are promising, but the evaluation scope is narrow.**
Training runs are relatively short (<= 10 epochs) and rely on single seeds, and the evaluation focuses almost entirely on retrieval. Since retrieval gives only a partial view of how much of the teacher’s geometry the student retains, it would help to include a broader set of zero-shot classification benchmarks. These are cheap to compute because they do not require training, so even a small set of teacher–student pairs evaluated on a standard dataset such as ImageNet would already clarify the semantic gap between them and offer a direct comparison to the retrieval gap. This would also provide a more interpretable proxy for the geometric mismatch implied by the current results. In addition, a qualitative view of the embedding spaces, for example through dimensionality reduction of teacher and student representations on the same samples, could help identify which structural components the TE surrogates capture and which ones they miss.

**Requested Changes:**

## Conceptual clarity

**Unify the meaning of time in the TE formulation.**
Section 4.1 treats t as an optimization step, while Section 4.2.1 treats batch index as if it provided temporal structure, even though batches are shuffled. Unifying these perspectives, or stating directly that TE1 and TE2 should be viewed as geometric surrogates rather than temporal approximations, would reduce confusion.

**Clarify the link between Theorem 1 and the surrogate losses.**
Theorem 1 connects TE to Jacobian cosines under a linear Gaussian assumption. TE1 and TE2 instead operate on differences between unrelated samples. A short justification of why these differences still capture the intended signal, or an explicit acknowledgment of the heuristic gap, would make the theoretical transition more coherent.

**Avoid phrasing that implies teacher dynamics.**
Expressions such as “directional changes of teacher and student” suggest that the teacher evolves over time, although it is fixed. Clarifying that variation arises across data rather than training steps would help.

## Method and training pipeline

**Provide an explicit description of a full training step.**
The current text does not fully specify how TE1 and TE2 integrate with the update rule or whether they require additional forward passes. A small pseudo-code block would make the procedure and its compute footprint clearer.

**Characterize the computational overhead relative to baselines.**
Appendix E explains why exact TE is intractable, but the cost of TE1 and TE2 relative to a CL-only or standard distillation run remains unclear. Approximate wall-clock comparisons or per-epoch timings for a few representative settings would strengthen the efficiency claim.

## Loss design and evaluation

**Motivate the combination of CL, KL, MSE, ICL and TE.**
The loss formulation currently appears exploratory. A brief explanation of why these losses are grouped together and what TE contributes beyond them would clarify the design choices.

**Improve the robustness of the empirical evaluation.**
Most reported results appear based on single runs. Including standard deviations **at least across samples** would make comparisons more reliable. Additional metrics such as MRR would provide more nuance, especially if paired with explicit sample counts for each evaluation.

**Broaden the zero-shot classification evaluation.**
The Food-101 result in the appendix is useful, but a standard benchmark such as ImageNet would give a clearer view of how much semantic structure the student retains. Zero-shot classification is inexpensive to compute and could be added for a few teacher–student pairs.

**Reflect on why the student remains far from the teacher.**
Although TE1 and TE2 improve retrieval, the gap to the teacher remains large across all settings. A short discussion of which geometric aspects the surrogates do not capture would be helpful.

**Discuss the relationship to mutual information.**
Since TE is conditional mutual information, readers may expect either MI-based baselines or a conceptual comparison. Even a short discussion would help place the method in context.

## Minor suggestions

**Citation style (Section 2.1).**
Some references use parentheses where a citation command is required (for example, “(Chen et al., 2023)”). Adjusting this would improve clarity.

**Scope of related work (Section 2.2).**
The video retrieval references feel tangential to the main topic. Removing or reducing this material would focus the discussion on structurally related methods.

**Explain the computational intractability of exact TE (Section 4.2).**
The text mentions a computational explosion but provides limited intuition. A short explanation of the underlying causes, such as high-dimensional density estimation and pairwise dependency scaling, would help readers understand the motivation without jumping to Appendix E.

**Clarify the logic behind TE2 (Section 4.2.2).**
Cosine similarity is nonlinear, so concatenating differences is not equivalent to pooling separate cosines. A concise geometric rationale or small empirical comparison would strengthen the reasoning.

**Hyperparameter explanation (Section 5.1).**
The statement about assigning larger weights to losses with smaller magnitudes is vague. Describing the heuristic more explicitly, or commenting on how the final values were chosen, would improve reproducibility.

**Architecture details (Section 5.2).**
While correct and helpful, these details are overly granular for the main text. Moving most of them to the appendix would streamline the exposition.

**Clarify the type of representations being distilled (Section 7).**
The conclusions state that TE focuses on global embedding statistics, which effectively means pooled representations rather than token-level features. Introducing this key aspect earlier would reduce ambiguity.

**Soften the claim of computational efficiency (Section 7).**
TE1 and TE2 are more efficient than exact TE, but it is unclear how they compare to standard distillation. Unless runtime measurements are added, the claim should be qualified.

**Clarify the setup in Table 4.**
Table 4 involves vision-only students, but it is not clear how the distillation loss is computed in this case or whether the text encoder contributes to TE. A short clarification would prevent confusion.

---

> ### Author Response · Authors · 2026-01-05
> **Conceptual clarity, method and training pipeline, loss design and evaluation**
>
> We are grateful for the reviewer’s thorough and constructive comments. Below, we respond to the concerns and describe the corresponding revisions.
>
> **Clear empirical gains, but the conceptual connection to TE needs sharpening.** In the revision, we have unified the definition: in Section 4.1, $t$ always denotes the optimization step, and the TE term is defined across updates ($t\rightarrow t+1$) on the same replayed mini-batch. Correspondingly, in Section 4.2 we clarify that the indices in TE1/TE2 do not assume any temporal structure from shuffled batch ordering; rather, TE1/TE2 are geometric finite-difference surrogates intended to capture the same cross-step signal motivated by Theorem 1 (Jacobian-alignment): we measure whether the student’s representation change induced by the update aligns with the teacher-implied change, using cached/replayed features. We remove/replace any text implying that “consecutive elements in a batch approximate time,” and we revise Sections 4.1 and 4.2 to explicitly state that TE1/TE2 should be interpreted as efficient geometric proxies for cross-step teacher$\rightarrow$student influence, not as temporal TE estimated from shuffled batches.
>
> **The interpretation of TE shifts throughout the paper.** We have revised Sections 4.1 and 4.2: (i) unify notation so $t$ denotes optimization steps, (ii) remove phrasing that implies teacher dynamics, and (iii) state explicitly that TE1/TE2 should be interpreted as Jacobian/geometry alignment proxies (motivated by Theorem 1) rather than temporal approximations from shuffled batches.
>
> **The experiments are promising, but the evaluation scope is narrow.** In the revision, we have: (i) added Section 5.3 to strengthen robustness by running 5 independent random seeds and reporting mean$\pm$ std for all retrieval metrics in Tables 8 and 9, and (ii) added cross-dataset zero-shot classification on ImageNet-1k in Appendix H.4. The dimensionality-reduction visualization is provided in Appendix H.5.
>
> ****Requested Changes:****
>
> **Conceptual clarity**
>
> **Q1** In the revision, we have made $t$ denote optimization step throughout, and we explicitly state that TE1/TE2 are geometric surrogates for teacher–student update alignment (Jacobian/finite-difference alignment), not temporal approximations over shuffled batch order. Concretely, in Sec. 4.2.1 we remove language implying that batch index provides temporal structure and clarify that indices (e.g., $i$, $i+1$) refer to consecutive optimization iterations on the same mini-batch via a lightweight replay/feature buffer. We revise Sec. 4.1 and Sec. 4.2 to unify notation and avoid confusion.
>
> **Q2** TE1/TE2 compute cosine similarity between within-batch feature differences. We do not interpret the batch index as time. Instead, these differences provide a finite-difference probe of local geometry in representation space; the student and teacher are evaluated on the same mini-batch, and we measure whether the student’s local variation aligns with the teacher’s. This is a heuristic motivated by Theorem 1 (Jacobian-alignment signal), and we clarify in Sec. 4.2 that TE1/TE2 are geometric surrogates rather than temporal TE estimates.
>
> **Q3** We have revised the text in Section 4.2.1 to avoid implying that the teacher evolves over time.
>
> **Method and training pipeline**
>
> **Q1** In Section 4.4, we have added a pseudo-code block.
>
> **Q2** We have added Section 5.3 (1st and 2nd paragraphs and Table 7) to address this.
>
> **Loss design and evaluation**
>
> **Q1** We have included 1st and 2nd paragraphs in Section 4.3 for explanation and clarification.
>
> **Q2** We have added Section 5.3 to address the robustness of the empirical evaluation.
>
> **Q3** We have added Appendix H.4 for zero-shot classification of ImageNet. The student model RN34 is distilled from RN50 based on MSCOCO and Food-101.
>
> **Q4** We have discussed it in Section 7.
>
> **Q5** We have added mutual information (MI)-based approach (Appendix F.6 and Section 4.3, $\eta>0$ in eqn (19)) and have compared it with the TE-based distillation in all experiments.
>
> **Minor suggestions**
>
> **S1** The references have been adjusted
>
> **S2** The video-retrieval references and discussions have been removed.
>
> **S3** We have explained it in the first paragraph of Section 4.2.
>
> **S4** We have added a paragraph: Why concatenation is meaningful (and not equivalent to averaging), in Section 4.2.2, as a geometric rationale.
>
> **S5** We have added a paragraph: Hyperparameter selection and reproducibility, at the end of Section 5.1.
>
> **S6** We have switched the architecture details to Appendix G.
>
> **S7** We have added this key aspect at the end of Section 4.2.
>
> **S8** We have added Section 5.3 to provide more details on the runtime measurements.
>
> **S9** In all experiments in the paper, including Table 4, the student is a full vision–language model with both an image encoder and a text encoder. We have updated the captions of all tables and figures to make this clearer and avoid confusion.

---

> > ### Comment · Action_Editor_mTSE · 2026-02-09
> > **Follow-up on Algorithm 1: Clarifying the Relationship Between Eq. (3), Eq. (5), and the Mixed Perturbation Design**
> >
> > Hi Authors,
> >
> > Thank you for the detailed rebuttal and the substantial revision of the manuscript, including the added Algorithm 1 and the expanded explanations. We appreciate the effort invested in clarifying the method and strengthening the paper.
> >
> > The newly added pseudo-code block in section 4.4 (i.e. Algorithm 1) clarifies the implementation of the proposed method, but it also exposes a conceptual issue that would benefit from explicit discussion in the paper.
> >
> > Eq. (3) defines transfer entropy across optimization steps, i.e., with state transitions induced by parameter updates between successive training iterations—an interpretation explicitly emphasized in the last two paragraphs of Section 4.1. In contrast, Eq. (5), together with the construction in Appendix C, corresponds to a Jacobian-cosine–based geometric surrogate defined under an input-perturbation setting, and is **not** the TE defined in Eq. (3). Algorithm 1 combines these two formulations by using parameter-induced representation changes on the student side and input-difference–based signals on the teacher side. More explicitly, Algorithm 1 is non-symmetric in that the student-side signal is computed from the **same samples across successive optimization steps**, whereas the teacher-side signal is computed from **different samples on the same parameter**; it would be helpful to clarify the rationale for aligning these two types of signals.
> >
> > To avoid ambiguity for readers, it would be helpful for the authors to clarify how Eq. (3), Eq. (5), and Algorithm 1 are intended to relate to one another. In particular, please clarify whether Eq. (3) and Eq. (5) are meant to represent the same underlying objective under additional assumptions, or whether they should be understood as conceptually distinct quantities, with Eq. (5) serving only as a heuristic surrogate. If an equivalence is intended, please explicitly state the assumptions under which this equivalence is claimed and explain how these assumptions are satisfied in Algorithm 1.
> >
> > More concretely, since Algorithm 1 applies parameter-induced perturbations on the student side and input-induced differences on the teacher side, a brief conceptual explanation (and any required assumptions) of why these signals are expected to be meaningfully aligned—beyond a purely heuristic regularizer—would help readers correctly interpret the method.
> >
> > Clarifying these points in the main text (e.g., around Section 4.1–4.2) would improve the conceptual coherence of the paper and reduce the risk of readers interpreting Eq. (5) as a direct approximation of the TE defined in Eq. (3).

---

> > > ### Author Response · Authors · 2026-02-12
> > > **Clarifications on Eqs. (3) and (5) and Updated Algorithm 1**
> > >
> > > We sincerely thank the Action Editor, Dr. Guo, for the careful reading and insightful comment on Algorithm 1 and the relationship between Eqs. (3) and (5).
> > >
> > > We agree with the core point: Eq. (3) defines an ideal teacher$\rightarrow$student transfer-entropy quantity across optimization steps (state transitions induced by parameter updates), whereas Eq. (5) is a Jacobian–cosine expression derived in an input-perturbation (local linearization) setting. These two quantities are not identical, and we do not intend Eq. (5) to be interpreted as a direct estimator of the optimization-step TE in Eq. (3).
> > >
> > > In the revision, we make this relationship explicit in Section 4.2 (immediately following Theorem 1). Specifically, we position Eq. (3) as a conceptual target where an information-flow diagnostic formalizing the notion that teacher signals should help explain the student’s next-step representations beyond the student’s current state, while Eq. (5) is used only to motivate a tractable geometric proxy via a first-order linear–Gaussian analysis (details in Appendix C). We also state clearly that our TE terms (TE1/TE2) are “TE proxies” (geometric surrogates) rather than estimates of the conditional mutual information in Eq. (3).
> > >
> > > Importantly, to address the potential ambiguity highlighted by the AE, we updated Algorithm 1 to match our actual experimental implementation and to remove the non-symmetric construction. In our implementation, both teacher- and student-side directional signals are instantiated via input-induced finite differences within the current mini-batch (paired examples after shuffling/permutation). TE1/TE2 then reward cosine alignment between teacher and student difference vectors (with TE2 using concatenated multimodal differences). This makes the intended role of Eq. (5) fully transparent: it provides theoretical motivation for using Jacobian-alignment–type geometric signals, while Algorithm 1 implements an efficient within-batch finite-difference approximation that transfers local representation geometry from teacher to student.
> > >
> > > We believe these clarifications and the updated pseudo-code of Algorithm 1 improve the conceptual coherence of Sections 4.1–4.2 and reduce the risk of readers interpreting Eq. (5) as a direct approximation of the TE defined in Eq. (3).

---

### Review · Reviewer_UpcZ · 2025-12-21

**Summary Of Contributions:**

This paper investigates knowledge distillation for CLIP-style vision-language models by introducing a regularization term based on Transfer Entropy (TE). The authors propose that TE can quantify the directional dependency between teacher and student embeddings, thereby facilitating a more structured knowledge transfer than standard feature mimicry. To make this information-theoretic concept computationally tractable, they derive a connection to Jacobian alignment under linear-Gaussian assumptions. This theory is ultimately relaxed into two practical surrogates, TE1 and TE2, which utilize the cosine similarity of embedding differences between adjacent samples in a batch. The effectiveness of the approach is evaluated on cross-modal retrieval tasks using the MSCOCO 2014 and Flickr8k datasets.

**Audience:**

Yes

**Audience Explanation:**

The paper explores an interesting conceptual shift from point-wise feature alignment to relational or directional dependency. By framing distillation through the lens of information flow, the authors attempt to address a fundamental challenge in VLM compression: preserving the intricate structure of the multimodal embedding space rather than just the final output values. This perspective is a welcome addition to the literature on model distillation.

From a practical standpoint, the proposed TE-VLM framework is highly efficient. The use of cosine similarity on embedding differences provides a lightweight regularization term that adds negligible computational overhead during training. It requires no modifications to the student architecture and integrates seamlessly into existing contrastive learning pipelines. This makes the method particularly attractive for resource-constrained environments where complex gradient-level alignments or large-scale teacher-student interactions are infeasible.

**Claims And Evidence:**

Yes

**Claims Explanation:**

The authors have put significant effort into the empirical analysis of their method. The ablation studies cover a wide range of hyperparameters and components, providing a clear picture of how different loss terms contribute to the final performance. Furthermore, the inclusion of a synthetic Gaussian channel experiment in Appendix D demonstrates a commendable effort to validate the mathematical foundations of the surrogate metrics under controlled, idealized conditions. The reported improvements in retrieval metrics across various student backbones, such as ResNet and ViT, suggest that the proposed heuristic captures some useful geometric signals that standard losses might miss.

**Requested Changes:**

1. Substantially reframe the theoretical section to acknowledge the heuristic nature of the surrogates. If the authors wish to maintain the Transfer Entropy narrative, they must provide a rigorous justification for why cross-sample differences in a shuffled, high-dimensional batch can serve as a proxy for same-sample Jacobian dynamics. Otherwise, the method should be re-characterized as a geometric alignment heuristic.
2. Include cross-dataset evaluation results. To demonstrate that the method actually improves the quality of the multimodal embedding space, the authors must show performance on datasets the model was not distilled on. This is essential for any claim regarding CLIP-style distillation.
3. Redesign Figure 1 to emphasize conceptual clarity. The figure should be simplified to show the specific mechanism of the TE loss and how it constrains the student’s representation relative to the teacher, removing the unnecessary raw data points.
4. Standardize and strengthen the baseline comparisons. Ensure every table includes a consistent set of baselines, including the strongest non-TE combination of existing losses. The authors should also confirm that their baseline performance matches the state-of-the-art for the architectures used.

---

> ### Author Response · Authors · 2026-01-05
> **Reframed Theory, Cross-Dataset Evaluation, Clearer Figure, and Stronger Baselines**
>
> We sincerely thank the reviewer for the detailed and constructive feedback. Our responses and revisions are summarized below.
>
> **Requested Changes**
>
> **Q1** We agree that our practical objectives are best stated in geometric terms, and we have revised Sections 4.1 and 4.2 accordingly. In our method, TE1/TE2 are not interpreted as estimating temporal transfer entropy from shuffled batches. The index $i$ enumerates examples within a mini-batch and does not represent time; consequently, cross-example differences should not be read as teacher/student dynamics. Instead, TE1/TE2 are used as geometric alignment surrogates: they measure whether the student's local representation variation on a batch aligns with the teacher's variation on the same batch. This design is heuristically motivated by Theorem 1, which links one-step TE under a local linear-Gaussian model to cosine similarity of (normalized) Jacobians; in high dimensions we replace explicit Jacobians with finite differences, using embedding differences as a practical proxy for Jacobian actions. We now explicitly acknowledge the remaining mismatch between the theorem's assumptions and the implemented surrogate, and we reframe TE1/TE2 as a Jacobian/geometry-inspired alignment heuristic rather than a rigorous estimator of transfer entropy in shuffled-batch settings. We have revised Sections 4.1 and 4.2 to unify the meaning of "time" (optimization step only) and to explicitly label TE1/TE2 as geometric surrogates motivated by Jacobian alignment, not temporal TE estimators.
>
> **Q2** We have added Appendix H.4: we distill an RN34 student from an RN50 teacher on MSCOCO and Food-101, and then evaluate cross-dataset generalization via zero-shot classification on the ImageNet-1k validation set (50K images) using the standard CLIP protocol. Importantly, the student is never trained on ImageNet, so this evaluation directly measures out-of-domain transfer and the quality of the learned multimodal embedding space under a substantial domain shift.
>
> **Q3** We have redesigned Fig. 1 to improve conceptual clarity by removing distracting elements and emphasizing the role of the TE term. Specifically, we reduced the number of subfigures (from 12 to 9) and removed the total-loss curves that did not directly explain the TE mechanism.
>
> **Q4** We have standardized and strengthened the baseline comparisons across the paper. In the revised manuscript, each results table (Tables 1, 3, 5, 11 and Tables 2, 4, 6, 12) now reports a consistent set of baselines, including the strongest non-TE configuration, e.g., mutual information-based approach $\eta>0$,  alongside our TE-regularized variants. We also verified that the best non-TE baselines are competitive with (and in our settings match) the expected performance for the corresponding CLIP teacher-student architecture under the same training data and evaluation setup.

---

### Decision · Action_Editor_mTSE · 2026-02-17

**Recommendation:** Reject

**Additional Comments:**

The authors have invested substantial effort in improving the manuscript. A major revision and resubmission could be considered if the next version more clearly aligns the framing, the final algorithmic specification, and the empirical validation.

**Audience:**

Yes

**Audience Explanation:**

This paper proposed a lightweight, plug-and-play geometry-based objective for CLIP-style distillation. The efficient distillation of multimodal models and simple objectives that can be added to existing pipelines are broadly relevant topics for the TMLR community.

**Claims And Evidence:**

No

**Claims Explanation:**

The authors have put substantial effort into the revision. The theory, algorithm description, and experiments have been expanded and clarified. However, based on the revised manuscript and the discussion, the current empirical evidence still does not fully and unambiguously support the paper’s high-level claims. In particular:

1. The mapping between the paper’s high-level positioning and what is actually implemented/optimized remains unclear.
2. The current experimental evidence is not strong enough to compensate for this gap.

As a result, the manuscript does not yet establish a clear, end-to-end link between its stated claims, the implemented objective, and the reported empirical validation.

### **1. Method Positioning and Implementation**

This section first summarizes the theoretical and algorithmic structure of the paper, and then explains why problems remain at the implementation level.

#### **Context for What the Paper Is Claiming**

This paper introduces the optimization-step transfer entropy in Eq. (3) as a high-level framing. In practice, the training objective is realized through a multi-stage proxy construction:
- from the optimization-step TE framing in Eq. (3),
- to the input-perturbation-based TE constructed in Appendix C (Eq. (24)),
- to the Jacobian–cosine proxy in Theorem 1 (Eq. (5)),
- and finally to within-batch finite differences implemented in Algorithm 1

Using a tractable proxy for an intractable ideal quantity is reasonable, but it requires clear scoping of what is conceptual framing versus what is actually optimized. In the rebuttal, the authors clarify that Eq. (5) is not equivalent to Eq. (3) (nor a direct estimator of it), and that TE1/TE2 are proxies; this distinction is now stated more clearly in Sections 4.1–4.2. However, the claim-level framing (e.g., Abstract/Introduction) can still reasonably be read as validating TE in the sense of Eq. (3), which goes beyond what the experiments directly test.

Accordingly, it would be important for the manuscript to state explicitly—at the claim level—that the reported experiments primarily validate the final proxy objective implemented in Algorithm 1, rather than a direct instantiation of the optimization-step TE framing in Eq. (3). While the revision improves this distinction in the technical sections, an equally explicit scope statement at the claim level would further reduce potential ambiguity in how readers map the claims to the empirical validation.

### **Concerns with Algorithmic Implementation**

On the algorithmic side, to address an earlier reviewer concern that the paper did not fully specify how TE1/TE2 integrate with the update rule and whether additional forward passes are required, the authors added Algorithm 1 to clarify the training procedure. This was a reasonable step. However, Algorithm 1 was later revised in a non-trivial way, and the manuscript states that the current version reflects the experimental implementation.

When an algorithm definition changes during the review/revision process, the manuscript should clearly state whether all reported results correspond to the final implementation. The current version does not explicitly confirm this. This creates uncertainty about the consistency between the algorithm description and the reported experimental results, and may raise reproducibility concerns.

### **2. Experimental Design and Evidence**

Since the method is explicitly positioned as a proxy/heuristic rather than a direct realization of Eq. (3), the evidence-to-claim mapping needs to be particularly clear and robust.

The paper reports that KL and MSE do not clearly decrease during training. This behavior is not analyzed in detail, and no recipe-level ablation (e.g., key hyperparameter diagnostics) is provided. The study also focuses heavily on loss-weight tuning while keeping key recipe-level parameters (e.g., temperature and batch size/negative pool strength) fixed without diagnostics, which is especially concerning given the KL/MSE behavior. Therefore, it is unclear whether the baseline configuration is well-behaved, and whether the reported gains can be confidently attributed to the proposed regularizer.

In addition, the paper does not include cross-dataset retrieval experiments (e.g., training on COCO and evaluating on Flickr, and vice versa), nor mixed-dataset training settings that more closely reflect CLIP-style training regimes. These experiments would better test the claims about structural knowledge preservation and generalization. The mechanism analysis (e.g., Figure 1) is also limited and does not clearly demonstrate how the proposed regularizer changes representation structure or training dynamics.

**Resubmission Of Major Revision:**

The authors may consider submitting a major revision at a later time.